# An essential periplasmic protein coordinates lipid trafficking and is required for asymmetric polar growth in mycobacteria

**Kuldeepkumar R Gupta[1†], Celena M Gwin[1†], Kathryn C Rahlwes[2], Kyle J Biegas[3,4], Chunyan Wang[1,5], Jin Ho Park[1], Jun Liu[1,5], Benjamin M Swarts[3,4], Yasu S Morita[2], E Hesper Rego[1]***

[1]Department of Microbial Pathogenesis, Yale University School of Medicine, New Haven, United States; [2]Department of Microbiology, University of Massachusetts, Amherst, United States; [3]Department of Chemistry and Biochemistry, Central Michigan University, Mount Pleasant, United States; [4]Biochemistry, Cell, and Molecular Biology Program, Central Michigan University, Mount Pleasant, United States; [5]Microbial Sciences Institute, Yale University, West Haven, United States

**\*For correspondence:**
hesper.rego@yale.edu

[†]These authors contributed equally to this work

**Competing interest:** The authors declare that no competing interests exist.

**Abstract** Mycobacteria, including the human pathogen *Mycobacterium tuberculosis*, grow by inserting new cell wall material at their poles. This process and that of division are asymmetric, producing a phenotypically heterogeneous population of cells that respond non-uniformly to stress (Aldridge et al., 2012; Rego et al., 2017). Surprisingly, deletion of a single gene – *lamA* – leads to more symmetry, and to a population of cells that is more uniformly killed by antibiotics (Rego et al., 2017). How does LamA create asymmetry? Here, using a combination of quantitative time-lapse imaging, bacterial genetics, and lipid profiling, we find that LamA recruits essential proteins involved in cell wall synthesis to one side of the cell – the old pole. One of these proteins, MSMEG_0317, here renamed PgfA, was of unknown function. We show that PgfA is a periplasmic protein that interacts with MmpL3, an essential transporter that flips mycolic acids in the form of trehalose mono-mycolate (TMM), across the plasma membrane. PgfA interacts with a TMM analog suggesting a direct role in TMM transport. Yet our data point to a broader function as well, as cells with altered PgfA levels have differences in the abundance of other lipids and are differentially reliant on those lipids for survival. Overexpression of PgfA, but not MmpL3, restores growth at the old poles in cells missing *lamA*. Together, our results suggest that PgfA is a key determinant of polar growth and cell envelope composition in mycobacteria, and that the LamA-mediated recruitment of this protein to one side of the cell is a required step in the establishment of cellular asymmetry.

## Editor's evaluation

This manuscript tackles the important and fundamental question of what proteins regulate asymmetrical cell division in Mycobacteria. This strong study will be of interest to all individuals interested in bacterial physiology and the conclusions are well-supported by the data.

## Introduction

*Mycobacterium tuberculosis*, the etiological agent of human tuberculosis (TB), is responsible for approximately 1.4 million deaths each year. One of the pathogen's distinguishing features is its unusual

cell envelope. Like nearly all other bacterial species, the plasma membrane is surrounded by peptido-glycan, a rigid mesh-like structure made up of carbohydrate chains cross-linked by peptide bridges. However, in contrast to the peptidoglycan of other well-characterized bacteria, mycobacterial peptidoglycan is covalently linked to the highly branched hetero-polysaccharide arabinogalactan, which is, itself, covalently bound to extremely long-chained fatty acids called mycolic acids. Collectively, this structure is known as the mycolyl-arababinogalactan-peptidoglycan complex or mAGP. Electron microscopy has revealed that the outermost layer is a lipid bilayer, and, as such, it is referred to as the outer membrane, or, alternatively, the mycomembrane (*Hoffmann et al., 2008*; *Zuber et al., 2008*). In addition to this core structure, several lipids, lipoglycans, and glycolipids are abundantly and non-covalently interspersed across the plasma membrane and mycomembrane (*Jackson, 2014*; *Jankute et al., 2015*). This complex cell envelope is a double-edged sword: it is a formidable barrier to many antibiotics yet provides several potentially targetable structures. Indeed, two of the four first-line TB antibiotics target cell envelope biosynthesis.

In addition to their unusual cell envelope, mycobacteria differ from other well-studied rod-shaped bacteria in important ways. Notably, mycobacteria elongate by adding new material at their poles rather than along their side walls. Over the course of a cell cycle, one pole grows more than the other, giving rise to an asymmetric growth pattern (*Aldridge et al., 2012*). Importantly, closely related organisms like corynebacteria that have similar cell wall architecture grow more evenly from their poles, suggesting that asymmetry is not simply a consequence of polar growth and may be actively created by mycobacteria (*Rego et al., 2017*). In fact, while the molecular details of asymmetric polar growth are not yet well understood (*Baranowski et al., 2019*; *Kieser and Rubin, 2014*), we have discovered that LamA, a protein of unknown function specific to the mycobacterial genus, is involved (*Rego et al., 2017*). Deletion of *lamA* results in more growth from the pole formed from the previous round of division, 'the new pole', and less from the established growth pole, 'the old pole', leading to less asymmetry (*Figure 1A*; *Rego et al., 2017*).

To understand LamA's role in mycobacterial division and elongation, we identified multiple putative LamA-interacting proteins of known and unknown function (*Rego et al., 2017*). One of these proteins, MSMEG_0317, is predicted to be associated with several other divisome proteins (*Wu et al., 2018*; *Figure 1A*). Attempts to delete *msmeg_0317* from the *Mycobacterium smegmatis* chromosome have been unsuccessful (*Cashmore et al., 2017*), and transposon insertion mapping has predicted the *M. tuberculosis* homolog, *rv0227c*, to be essential (*DeJesus et al., 2017*; *Zhang et al., 2012*). MSMEG_0317 belongs to the DUF3068-domain super-family of proteins, which are exclusively found in actinobacteria. One member of this family in corynebacteria has channel activity, so this domain has been renamed PorA (*Abdali et al., 2018*; *Soltan Mohammadi et al., 2013*). In addition, by structure-based homology prediction, MSMEG_0317 shares limited homology to CD36, which transports long-chained fatty acids into eukaryotic cells (*Patel et al., 2022*). In corynebacteria, the putative homolog of MSMEG_0317, LmcA, is thought to have lipid binding activity (*Patel et al., 2022*), and is involved in an ill-defined step of lipoglycan synthesis (*Cashmore et al., 2017*; *Patel et al., 2022*).

Here, we sought to understand the role of MSMEG_0317 in relation to LamA during mycobacterial growth and division. We find that in *M. smegmatis*, a model mycobacterial species, MSMEG_0317 is essential for polar growth, and localizes to the old pole in a LamA-dependent manner. There, it localizes with MmpL3, the essential mycolic acid flippase, to build the mycomembrane. Our data suggest that one function of MSMEG_0317 is to traffic mycolic acids in the periplasmic space, and that its over-expression is sufficient to restore growth at the old pole in cells missing *lamA*. Together, these results implicate MSMEG_0317 in mycolic acid trafficking, and argue that MSMEG_0317 is an important factor in determining cell wall composition and incorporation at one site of growth – the old pole – in mycobacteria. As such, we propose to rename MSMEG_0317, PgfA, for <u>P</u>olar <u>g</u>rowth <u>f</u>actor <u>A</u>.

## Results

### PgfA is essential for polar growth and localizes to the sites of new cell wall synthesis

To verify the essentiality of *pgfA* in *M. smegmatis*, we used an allele swapping strategy (*Pashley and Parish, 2003*). Briefly, in a strain whose only copy of *pgfA* was at the L5 phage integration site (*Lewis and Hatfull, 2000*; *van Kessel and Hatfull, 2007*), we exchanged *pgfA* for either an empty vector

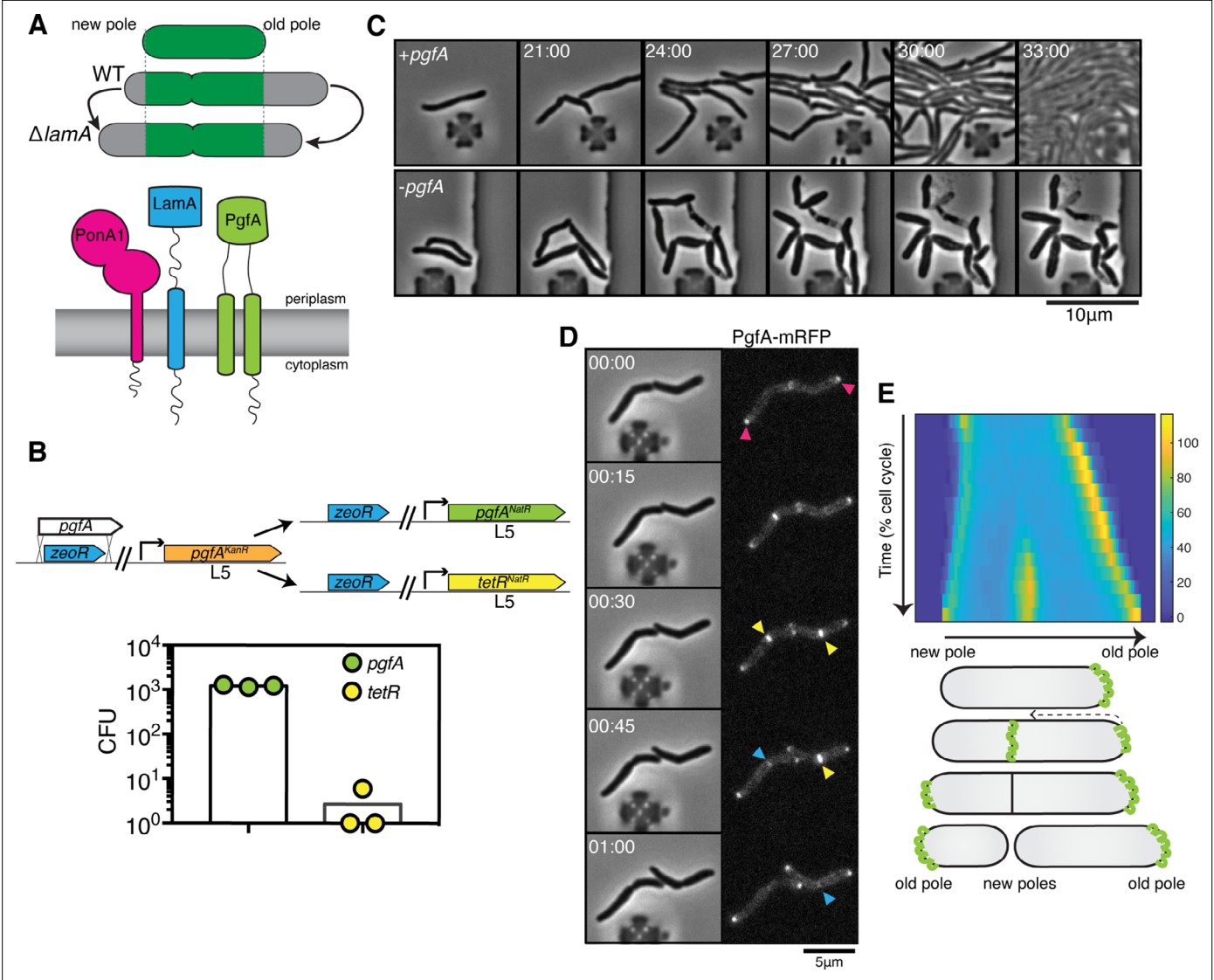

**Figure 1.** PgfA is an essential polar growth factor that localizes asymmetrically. (**A**) *Top:* Graphical depiction of growth pattern in wild type (WT) and Δ*lamA* cells. Green = old cell wall material. Gray = new cell wall material. *Bottom:* LamA is a membrane protein that co-immunoprecipitates with PonA1, a bifunctional penicillin binding protein, and MSMEG_0317/PgfA, a protein of unknown function. (**B**) Schematic and results of allelic exchange experiment. Vectors with *pgfA* or without *pgfA* (*tetR*) were transformed into a strain whose only copy of *pgfA* was at the L5 integration site. Transformants carrying the incoming vectors were counted by colony forming units (CFU). (**C**) A strain whose only copy of PgfA is tetracycline inducible was imaged over time with (+pgfA) or without (-pgfA) anhydrotetracycline (ATC). Cells were loaded into a microfluidic device 18 hr after the removal of ATC (bottom) or a mock control (top). (**D**) Cells whose only copy of PgfA was fused to mRFP were imaged over time by phase and fluorescence microscopy in a microfluidic device with constant perfusion of media. (**E**) *Top*: Individual cells (N=25) were followed from birth to division and the fluorescence was measured from new to old pole. Each resulting kymograph was interpolated over cell length and time and then averaged together. Using this analysis, we find that PgfA is first at the old poles (pink triangles), partially re-localizes to the septum (yellow triangles) during cell division, and then disappears from the site of division before the next cell cycle (blue triangles) to establish asymmetry in the next generation. *Bottom*: A depiction of this localization pattern is shown as a cartoon.

The online version of this article includes the following figure supplement(s) for figure 1:

**Figure supplement 1.** PgfA is essential and depletion leads to reductive cell division.

**Figure supplement 2.** Cells depleted of PgfA and MmpL3 do not grow from their poles.

**Figure supplement 3.** PgfA localizes to midcell and the poles.

or for another copy of itself (*Figure 1B*). Consistent with *pgfA* being essential for cell growth, we observed approximately 1000-fold fewer colonies when we exchanged *pgfA* with an empty vector compared to exchanging it for another copy of itself (*Figure 1B*).

To visualize the morphology of *pgfA*-depleted cells, we constructed a strain in which the only copy of the gene was tetracycline inducible (*Figure 1C*; *Figure 1—figure supplement 1A*). Removal of anhydrotetracycline (ATC) from the culture media prevented cell growth (*Figure 1—figure supplement 1A*). By time-lapse microscopy we observed that, while cells expressing PgfA became longer, on average, as cell density increased within the microfluidic device (*Figure 1C*; *Figure 1—figure supplement 1B*) cells depleted for PgfA stopped elongating but continued to divide (reductive division), became wider, and, in many instances, eventually lysed (*Figure 1C*, *Figure 1—figure supplement 1B*). As cells stopped elongating, we reasoned that PgfA is important for polar growth. To test this, we stained PgfA-depleted cells with a dye that is incorporated into peptidoglycan and monitored outgrowth in dye-free media. Consistent with the notion that PgfA is important for polar growth, we find that PgfA-depleted cells incorporate less new cell material at their poles (*Figure 1—figure supplement 2*).

If PgfA is important for polar growth, then it should localize to the poles. To determine where and when PgfA functions in the cell, we fused PgfA to mRFP and expressed the resulting chimera from the native *pgfA* promoter. By allele swapping, we find that P$_{native}$-*pgfA-mrfp* restores bacterial growth (*Figure 1—figure supplement 1C*) and thus encodes a functional PgfA. Fluorescence microscopy at a single timepoint showed that PgfA-mRFP localizes to mid-cell and to the poles (*Figure 1D*, *Figure 1—figure supplement 3A*). In polar growing bacteria, the site of division eventually becomes the site of elongation. Thus, in addition to localizing to the poles, elongation-complex proteins can also appear at mid-cell before daughter cell separation is clearly observed. This makes it difficult to determine true septal-associated localization from a single timepoint. To disentangle whether PgfA, in addition to localizing to the poles, also localizes to the septum during division, we visualized PgfA-mRFP by time-lapse microscopy (*Figure 1D*). For a single cell, we measured the fluorescence distribution over time as a function of both cell cycle time and cell length. As we wanted to compare across strains, we averaged many of these individual trajectories together. The resulting 'average' kymograph represents the probability of finding a fluorescent protein in a particular cellular location at a particular stage of the cell cycle (*Figure 1E*). Using this analysis, we compared the spatiotemporal localization of PgfA-mRFP to the earliest known markers for the division complex (FtsZ-mCherry2B) and the elongation complex (eGFP-Wag31) (*Figure 1—figure supplement 3B*). We find that PgfA localizes primarily to the old pole before the onset of division. During division, the fluorescence intensity at the old pole becomes slightly less intense as PgfA-mRFP re-localizes to the septum (*Figure 1D and E*). This event occurs during the latter stages of FtsZ recruitment, but before the arrival of Wag31 to the mid-cell, suggesting that PgfA is a late divisome-associated protein (*Figure 1—figure supplement 3*). At the end of division, PgfA-mRFP disappears from the septal site such that the new daughter cells are once again born with an asymmetric distribution of PgfA (*Figure 1D and E*). Taken together, these data are consistent with PgfA being a member of both the mycobacterial division and elongation complexes and show that PgfA is essential for polar growth.

## PgfA and MmpL3 are recruited to the old pole by LamA to build the mycomembrane

The manner of cell death suggested a defect in the cell envelope of PgfA-depleted cells. To resolve the layers of the mycobacterial cell envelope in detail we used cryo-electron microscopy (cryo-EM) to visualize frozen-hydrated cells. As has been previously observed (*Hoffmann et al., 2008*; *Zuber et al., 2008*), wild type *M. smegmatis* cells exhibit distinct plasma and outer membranes, both clearly observed as bilayers, separated by approximately 50 nm (*Figure 2A*). In cells depleted of PgfA, the outer membrane is frayed (*Figure 2A*; *Figure 2—figure supplement 1*) and largely devoid of electron density (*Figure 2A*). Thus, PgfA is important for maintaining the structural organization of mycomembrane.

Many of the cytoplasmic and periplasmic enzymes involved in mycomembrane synthesis have been identified. However, the molecular details of how precursors are trafficked from the plasma membrane to build the complex structure are almost completely unknown. We do know that one key step involves the essential protein MmpL3, which transports trehalose monomycolate (TMM) across

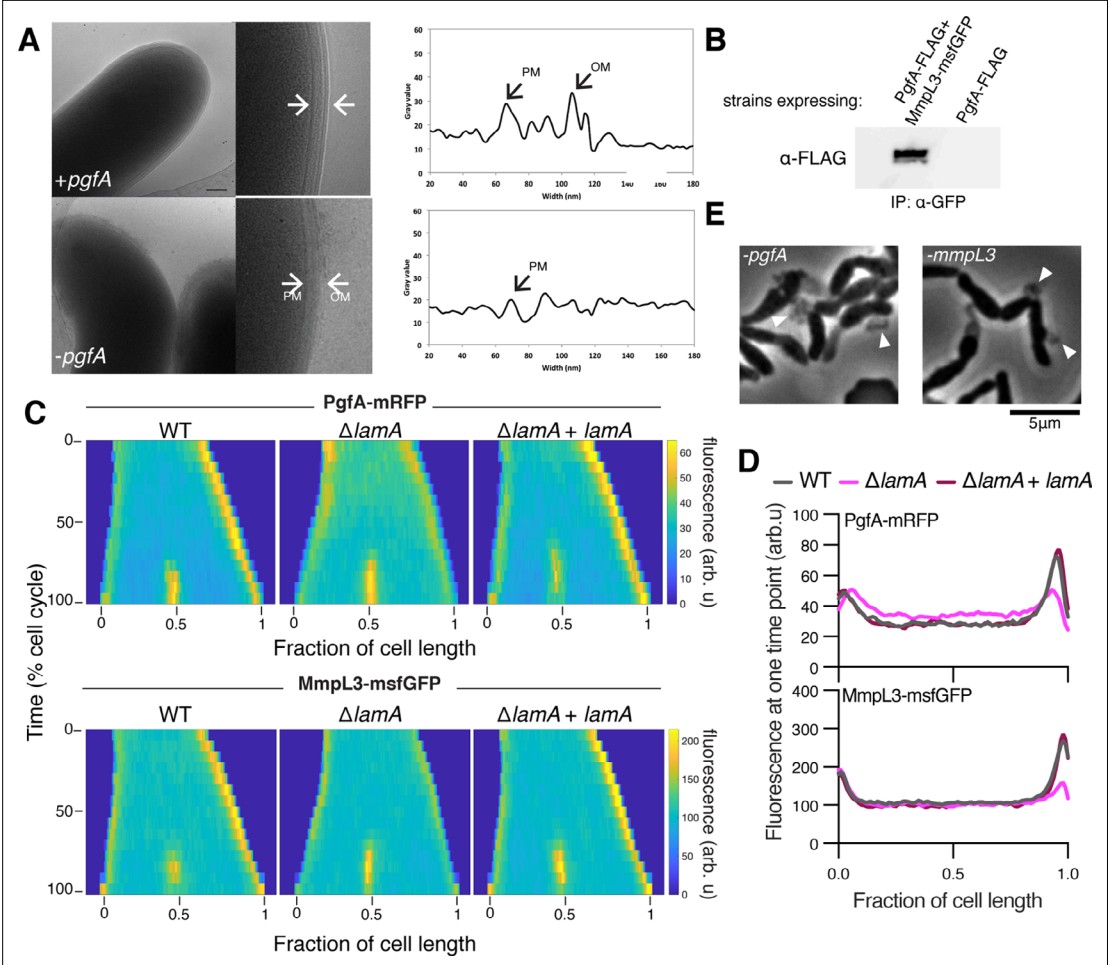

**Figure 2.** LamA recruits PgfA and MmpL3 to the old pole to build the mycomembrane. (**A**) The cell envelope of a representative wild type Msm cell and a cell depleted of PgfA for 24 hr were visualized and measured by cryo-electron microscopy. PM = plasma membrane; OM = outer membrane. Scale bar = 100 nm. (**B**) The lysates from strains expressing PgfA-FLAG with or without MmpL3-msfGFP were immunoprecipitated (IP) using GFP-trap beads, and the elution was probed for the presence of PgfA-FLAG via anti-FLAG Western blot. (**C**) Fluorescent protein fusions to either PgfA or MmpL3 were imaged by time-lapse microscopy in wild type, ΔlamA, and complement cells (ΔlamA +pNative lamA) N=25. The resulting images were then processed as described in *Figure 1D*. (**D**) The same data as panel C are plotted at a single timepoint to show the decrease in old pole accumulation of MmpL3 and PgfA in the absence of *lamA*. (**E**) Cells were depleted using CRISPRi for *pgfA* or *mmpL3* and a representative timepoint is shown. Cell wall material is excreted from the poles and septa during depletion (white triangles).

The online version of this article includes the following video, source data, and figure supplement(s) for figure 2:

**Source data 1.** Unprocessed anti-FLAG Western blot.

**Source data 2.** Unprocessed and labeled anti-FLAG Western blot.

**Figure supplement 1.** The cell envelopes of PgfA-depleted show outer membrane wall fraying at 24 hr (left) and 33 hr (right) of depletion.

**Figure supplement 2.** Full blots for the experiment in *Figure 2B*.

**Figure supplement 2—source data 1.** Unprocessed anti-FLAG Western blot.

**Figure supplement 2—source data 2.** Unprocessed and labeled anti-FLAG Western blot.

**Figure supplement 2—source data 3.** Unprocessed and labeled anti-GFP Western blot.

**Figure supplement 2—source data 4.** Unprocessed and labeled anti-GFP Western blot.

**Figure supplement 3.** Kymographs of a fluorescent fusion to MurJ in wild type (WT) and in ΔlamA cells.

**Figure supplement 4.** An anti-strep Western blot of wild type (WT) Msm and Msm carrying an extra copy of MSMEG_0315-strep.

**Figure supplement 4—source data 1.** Unprocessed anti-strep Western blot.

**Figure supplement 4—source data 2.** Unprocessed and labeled anti-strep Western blot.

**Figure 2—animation 1.** Phase time-lapse microscopy of cells carrying anhydrotetracycline (ATC)-inducible CRISRPi guides targeting either pgfA (left) or mmpL3 (right).

the plasma membrane to the periplasm (*Su et al., 2021*; *Xu et al., 2017*). There, TMM is trafficked, through unknown mechanisms, to the antigen 85 enzymes (*Backus et al., 2014*), and incorporated onto mAGP or made into trehalose dimycolate (TDM), a component of the outer mycomembrane leaflet (*Chiaradia et al., 2017*). Importantly, MmpL3 has become an anti-TB therapeutic target, as multiple compounds have been found to inhibit its function (*Adams et al., 2021*; *La Rosa et al., 2012*; *Umare et al., 2021*).

Using an *Escherichia coli* two-hybrid approach, MmpL3 from *M. tuberculosis* was recently found to interact with several factors involved in cell wall synthesis, growth, and division, including Rv0227c (64% identity to *M. smegmatis* PgfA) (*Belardinelli et al., 2019*). To verify that PgfA and MmpL3 interact in intact mycobacterial cells, we constructed strains that expressed fusions to these proteins to enable co-immunoprecipitation. Precipitating MmpL3-GFP with a nanobody against GFP, we found PgfA-FLAG in the elution only when MmpL3-GFP was present (*Figure 2B*; *Figure 2—figure supplement 2*). Taken together, these results show that MmpL3 and PgfA interact in the cell.

As our initial interest in PgfA was prompted by its connection to LamA, we wondered if PgfA and MmpL3 would localize differently in Δ*lamA* cells. Thus, we visualized fluorescent fusions to PgfA and MmpL3 in strains with and without *lamA* and analyzed the fluorescence distributions over time. MmpL3-msfGFP displayed the same spatial and temporal localization pattern as PgfA-mRFP, again suggesting they are part of the same complex (*Figure 2C and D*). Surprisingly, in Δ*lamA* cells, PgfA-mRFP and MmpL3-msfGFP became dramatically less abundant at the old pole, with no change in abundance at the new pole. The loss of polar PgfA and MmpL3 in Δ*lamA* could be complemented by integrating *lamA* at a single site on the chromosome (*Figure 2C and D*). The abundance of msfGFP-MurJ, which transports peptidoglycan precursors, was not as dramatically changed at the old pole in Δ*lamA* cells, showing that the LamA-dependent loss of MmpL3 and PgfA was specific and not a general loss of elongation-complex proteins (*Figure 2—figure supplement 3*).

If PgfA and MmpL3 are functioning together, then cells depleted of either essential protein may have the same terminal phenotype. To assay this, we created CRISPRi guides to both *mmpL3* and *pgfA* and visualized the morphology of cells over time as the depletion was induced (*Rock et al., 2017*). As *pgfA* is predicted to be the first gene in a two gene operon (*Martini et al., 2019*), we were concerned about polar effects of the CRISPRi depletion (*Rock et al., 2017*). To address this, we integrated another copy of MSMEG_0315, the second gene in the operon, at a phage site expressed by its native promoter, and confirmed its expression by Western blot (*Figure 2—figure supplement 4*). Consistent with the notion that PgfA and MmpL3 function in the same pathway, we found that cells depleted for *mmpL3* phenocopied those depleted for *pgfA* in that they become progressively shorter and wider (*Figure 2—animation 1*), and incorporated less material at their poles (*Figure 1—figure supplement 2*). By phase contrast microscopy, we also observed cell wall material excreted from the poles and the septa in both *mmpL3*- and *pgfA*-depleted cells (*Figure 2E*). These observations are reminiscent of cells treated with ethambutol (*Wuo et al., 2022*), a drug that inhibits the synthesis of arabinan, the anchor point for mycolic acids (*Kilburn and Takayama, 1981*; *Mikusová et al., 1995*). Collectively, our data are consistent with a model in which LamA recruits PgfA and MmpL3 to the old pole, either directly or indirectly, where they function to construct the mycomembrane.

## PgfA is a periplasmic protein that binds TMM

A recent proteomics study identified PgfA as a putative TMM-interacting protein (*Kavunja et al., 2020*). To verify this, we tested whether PgfA could be captured from live *M. smegmatis* cells using N-x-AlkTMM-C15, a synthetic photo-cross-linkable TMM analog containing an alkyne 'click' chemistry handle that enables specific detection and/or enrichment of cross-linked protein interactors (*Kavunja et al., 2020*). *M. smegmatis* cells expressing PgfA-3× FLAG were incubated with N-x-AlkTMM-C15 and exposed to UV irradiation to effect cross-linking. Then, cell lysates were collected and subjected to 'click' reaction to dual label N-x-AlkTMM-C15-modified proteins with fluorophore and biotin affinity tags. Biotinylated proteins were captured on avidin beads, eluted, and analyzed by anti-FLAG Western blot. We found that PgfA was enriched exclusively in N-x-AlkTMM-C15-treated, UV-exposed *M. smegmatis*, demonstrating direct interaction between PgfA and the TMM analog (*Figure 3A* and *Figure 3—figure supplement 1*).

Several accessory proteins work with MmpL3 and/or are required to transport TMM across the plasma membrane. However, to date, all the known accessory proteins reside in the cytoplasm or

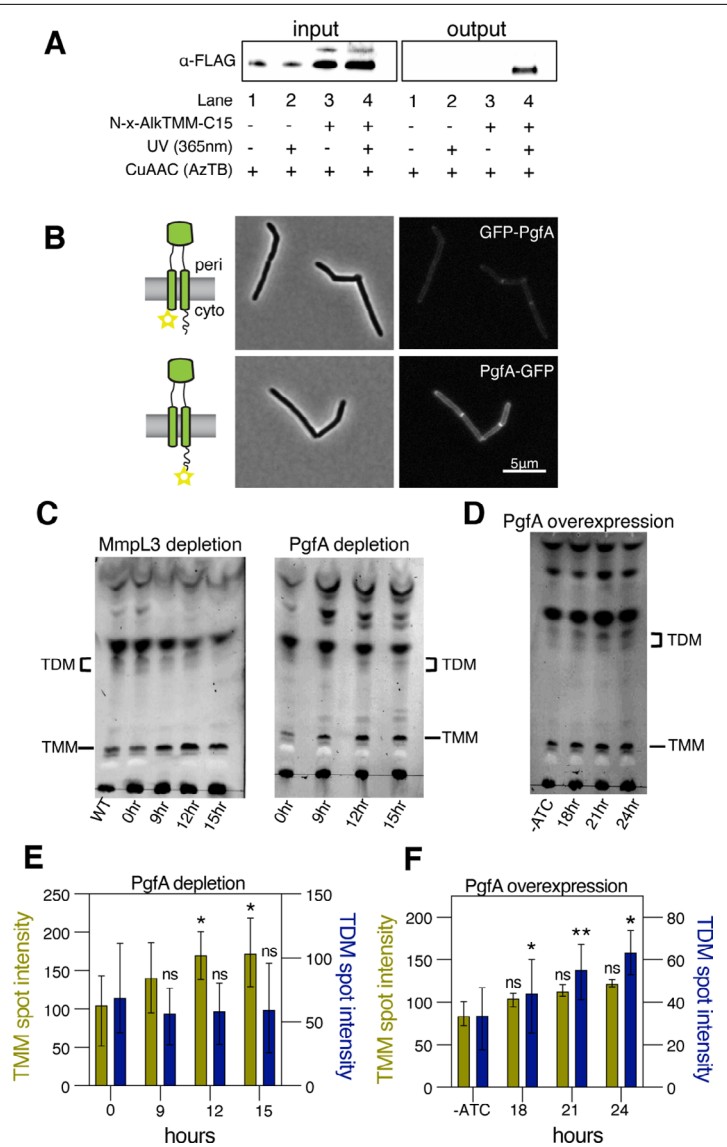

**Figure 3.** PgfA is a periplasmic protein that binds trehalose monomycolate (TMM) and is involved in TMM trafficking. (**A**) Cells expressing PgfA-3× FLAG were cultured with N-x-AlkTMM-C15 (100 μM), UV-irradiated, and lysed. Lysates were reacted with azido-TAMRA-biotin reagent (AzTB) by Cu-catalyzed azide-alkyne cycloaddition (CuAAC 'click' reaction), and analyzed before (input) and after (output) avidin bead enrichment by anti-FLAG Western blot. The full analysis, including Coomassie and in-gel fluorescence scanning, is displayed in **Figure 3—figure supplement 1**. Data are representative of two independent experiments. (**B**) PgfA was fused to GFPmut3 at either its N- or C-terminus and integrated into Msm as the sole copy of PgfA. Cells carrying these fusions were imaged by fluorescence microscopy. Images are displayed on identical intensity scales to allow direct comparison. (**C**) Using CRISPRi, cells were depleted of MmpL3 or PgfA at the indicated timepoints and the cell envelopes were analyzed for TMM and trehalose dimycolate (TDM) by thin-layer chromatography (TLC). (**D**) Cells were induced to overexpress PgfA at the indicated timepoints, and their cell envelopes were analyzed by TLC for TMM and TDM. (**E,F**) Quantification of three biological replicates analyzed by TLC as shown in panels C and D. To determine statistical significance, a paired one-way ANOVA was performed that compared the intensity at the indicated timepoints to the intensity at time 0 or a no ATC control within the same replicate. *p<0.05; **p<0.005.

The online version of this article includes the following source data and figure supplement(s) for figure 3:

**Source data 1.** Unprocessed MmpL3 depletion thin-layer chromatography (TLC).

**Source data 2.** Unprocessed and labeled MmpL3 depletion thin-layer chromatography (TLC).

**Source data 3.** Unprocessed PgfA depletion thin-layer chromatography (TLC).

*Figure 3 continued on next page*

*Figure 3 continued*

**Source data 4.** Unprocessed and labeled PgfA depletion thin-layer chromatography (TLC).

**Source data 5.** Unprocessed PgfA overexpression thin-layer chromatography (TLC).

**Source data 6.** Unprocessed and labeled PgfA overexpression thin-layer chromatography (TLC).

**Figure supplement 1.** Full results of experiment present in main text *Figure 3A*.

**Figure supplement 1—source data 1.** Unprocessed Coomassie gel.

**Figure supplement 1—source data 2.** Unprocessed in-gel fluorescence scan.

**Figure supplement 1—source data 3.** Unprocessed anti-FLAG Western blot.

**Figure supplement 1—source data 4.** Full uncropped gels with relevant bands labeled.

**Figure supplement 2.** PgfA is a transmembrane protein with two cytoplasmic tails.

**Figure supplement 3.** Control thin-layer chromatography (TLC) for TMM/TDM levels.

**Figure supplement 3—source data 1.** Unprocessed thin-layer chromatography (TLC) comparing wild type (WT) Msm to Msm Δ*antigen85A*.

**Figure supplement 3—source data 2.** Unprocessed and labeled thin-layer chromatography (TLC) comparing wild type (WT) Msm to Msm Δ*antigen85A*.

**Figure supplement 3—source data 3.** Unprocessed thin-layer chromatography (TLC) comparing wild type (WT) Msm +/-isoniazid.

**Figure supplement 3—source data 4.** Unprocessed thin-layer chromatography (TLC) and labeled comparing wild type (WT) Msm +/-isoniazid.

have globular domains on the cytoplasmic side of the plasma membrane (*Fay et al., 2019*). The topology of the predicted PgfA structure shows a beta-barrel-like domain anchored by one or, possibly two, transmembrane helices (*Patel et al., 2022*). Prediction of the orientation with respect to the membrane is ambiguous (*Figure 3—figure supplement 2A*). To map the topology of PgfA, we fused mGFPmut3 to either the N- or the C-terminal side of PgfA (*Figure 3B*). As GFP does not fluoresce in the mycobacterial periplasm (*Fay et al., 2019*), we reasoned that fluorescence would indicate cytoplasmic localization of GFP. Importantly, both fusion proteins can replace wild type PgfA at high efficiency, and thus encode functional PgfA (*Figure 3—figure supplement 2B*). Using fluorescence microscopy, we find that the C-terminal fusion is brightly fluorescent, supporting an orientation that places the beta-barrel-like domain in the periplasmic space. Interestingly, the N-terminal fusion is significantly dimmer (*Figure 3B*), suggesting that further experimentation will be needed to fully understand PgfA's interaction with the membrane. Nevertheless, collectively, these data show that PgfA is an inner membrane protein with a large periplasmic domain and that some portion of the protein binds TMM.

## PgfA abundance correlates to altered TMM/TDM ratios

If PgfA is involved in the trafficking of TMM, then its depletion, as with depletion of MmpL3, should result in an altered TMM/TDM ratio (*Degiacomi et al., 2017*; *Fay et al., 2019*; *Su et al., 2019*; *Tahlan et al., 2012*). Total cell-associated amounts of TMM and TDM can be visualized by thin-layer chromatography (TLC) (*Figure 3—figure supplement 3*). As expected, depletion of *mmpL3* leads to accumulation of TMM and a co-incident reduction of TDM (*Figure 3C*). To test if this is also the case for PgfA, we repeated the same procedure on cells depleted of *pgfA*. In this case, we find that TMM accumulates while TDM levels are more stable (*Figure 3C and E*). These data are consistent with a model in which TMM accumulates in a different subcellular compartment in *pgfA*-depleted cells, compared to *mmpL3*-depleted cells. Specifically, it suggests that in *pgfA*-depleted cells, at least some of the TMM pool may still be accessible to antigen 85 enzymes.

To further test the notion that PgfA is involved in TMM trafficking in the periplasm, we created a strain that inducibly overexpressed the protein from a multi-copy plasmid. Again, consistent with the notion that PgfA is involved in trafficking TMM, we find that inducing the overexpression of PgfA results in increased TDM levels (*Figure 3D and F*). Together these data support a model in which PgfA functions as part of the TMM transport pathway in the periplasm.

## The abundance of PgfA negatively correlates with the abundance of lipoglycans LM/LAM

Our data indicate a role for PgfA in the transport of TMM. In contrast, the presumed corynebacterial homolog of *pgfA* – named *lmcA* – has been proposed to function in the biosynthesis of two large lipoglycans abundant in the mycobacterial cell envelope, lipomannan and lipoarabinomannan (LM/LAM) (*Cashmore et al., 2017*). A deletion mutant of *lmcA* in *Corynebacterium glutamicum,* resulted in disappearance of LAM and accumulation of truncated LM, which was clearly detectable by faster migration on SDS-PAGE (*Cashmore et al., 2017*). To test if this is also the case in mycobacteria, we used the same electrophoretic approach to analyze LM/LAM, and TLC to examine potential changes in other lipid species upon depletion of *pgfA*. In contrast to the results obtained in *C. glutamicum*, we observe a dramatic and transient increase in the total amount of cell-associated LAM during depletion, before cells began dying (*Figure 4A*, *Figure 4—figure supplement 1*). There were no obvious changes in the migration of LM/LAM on SDS-PAGE, implying that, unlike deleting *lmcA* in *C. glutamicum*, depleting PgfA in *M. smegmatis* minimally impacts LM/LAM biosynthesis. Additionally, other lipids, including the precursor of LM/LAM biosynthesis such as AcPIM$_2$, show no reproducible change (*Figure 4B and C*; *Figure 4—figure supplement 1*).

These data led us to hypothesize that PgfA and LmcA do not have the same function, even though there is synteny in their chromosomal location. Indeed, PgfA and LmcA share only 21% sequence identity and there are marked differences in their structures (*Patel et al., 2022*), most noticeably a long C-terminal cytoplasmic tail that is critical to the function of PgfA but missing in LmcA (*Figure 4D*; *Figure 4—figure supplement 2A–B*). Thus, we wondered if LmcA can substitute for PgfA in *M. smegmatis*. To test this, we used our allele swapping strategy. We found that *lmcA*, even with PgfA's cytoplasmic tail, is unable to substitute for *pgfA* in *M. smegmatis* (*Figure 4D*; *Figure 4—figure supplement 2C*). Thus, while both *pgfA* and *lmcA* mutants have altered LM/LAM profiles, collectively, our data suggest that further studies are needed to confirm if they are true orthologs.

Why does depletion of PgfA result in an accumulation of LM/LAM? To investigate this, we decided to leverage Δ*lamA* cells, which naturally have less PgfA and MmpL3 (*Figure 2C*). PgfA and MmpL3 have been described as highly vulnerable drug targets, that is, a small change in their abundance causes growth arrest or cell death (*Bosch et al., 2021*). How do Δ*lamA* cells grow at a normal rate with fewer of these essential proteins? We reasoned that changes in gene essentiality in Δ*lamA* may uncover compensatory mechanisms that promote survival. Thus, we created transposon libraries in Δ*lamA* and wild type cells and compared insertions across the genomes. There were very few changes: aside from Δ*lamA* itself, the only gene that sustained significantly fewer insertions in Δ*lamA* across replicates was MSMEG_3187 or *sucT* (*Figure 4E*; *Figure 4—source data 1*). SucT succinylates arabinogalactan and LAM, a modification that changes the structural properties of the mycomembrane (*Palčeková et al., 2019*). These data suggest that in cells deleted for *lamA*, in which the levels of PgfA and MmpL3 are lower, the presence of lipoglycans or their modifications become more important. In fact, we find that cells overexpressing PgfA, which results in more TDM (*Figure 3D and F*), also have lower levels of cell-associated LAM (*Figure 4F*).

Taken together, our data suggest that PgfA is important for maintaining the correct levels of LM/LAM in the cell envelope. Further, our data uncover a previously unknown connection between TMM transport, or assembly of the mycomembrane, and LM/LAM. We find that the levels of LAM, and to a lesser extent LM, are negatively correlated with the levels of PgfA-mediated TMM trafficking, suggesting that there may be competing pathways for the transport or synthesis of these molecules, a notion supported genetically by Tn-seq. Alternatively, altered PgfA-mediated TMM trafficking may lead to differences in LM/LAM accessibility to extraction methods.

## PgfA is necessary and sufficient for old pole growth in Δ*lamA* and the molecular requirements for growth between the poles are different

We have shown that PgfA and MmpL3 are necessary for polar growth (*Figure 1C–E*; *Figure 1—figure supplement 2*). Additionally, in Δ*lamA* cells, which grow less from the old pole, PgfA and MmpL3 are less abundant at this site (*Figure 2C–D*). We wondered if loss of old pole growth in Δ*lamA* cells could be directly attributed to the lowered abundance of either PgfA or MmpL3 at this site. To test if PgfA and/or MmpL3 are sufficient to restore old pole growth in Δ*lamA*, we created strains that expressed a second copy of these genes driven by a strong ribosomal promoter. These strains had no discernible

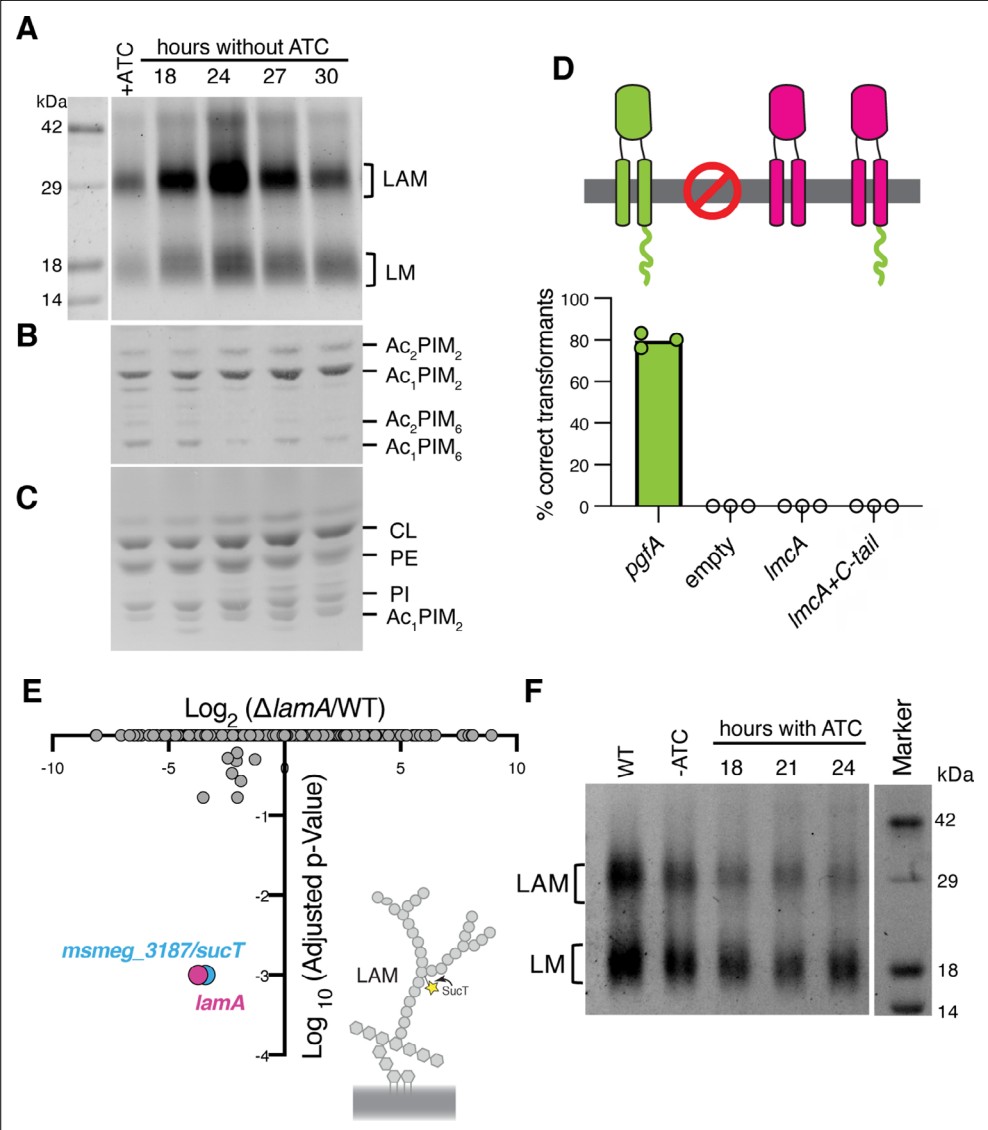

**Figure 4.** Lipomannan and lipoarabinomannan (LM/LAM) levels negatively correlate with abundance of PgfA. In a strain whose only copy of *pgfA* is anhydrotetracycline (ATC) inducible, (**A**) LM/LAM, (**B**) non-polar and polar PIMs, (**C**) and plasma membrane lipids were analyzed by (**A**) SDS-PAGE and (**B–C**) thin-layer chromatography (TLC) during PgfA depletion (without ATC). Cell pellets were normalized by wet weight to account for differences in cell density. To avoid changes in LM/LAM due to cell density, at the indicated timepoints, half of the cultures were taken for lipid analysis, and replaced with fresh media. Two other biological replicates are shown in *Figure 4—figure supplement 1*. CL, cardiolipin; PE, phosphatidylethanolamine; PI, phosphatidylinositol. (**D**) Results of exchanging the indicated *pgfA* and *Corynebacterium glutamicum NCgl2760* alleles for a wild type copy of *pgfA*. Correct transformants are positive for the incoming selective marker and negative for the outgoing (KanR, NatS). (**E**) Transposon insertions in Δ*lamA* compared to wild type (WT). The only two genes with significantly different number of insertions (reduced) were *lamA* itself and *sucT*, a gene that codes for a protein known to modify LAM and arabinogalactan. (**F**) In WT cells and those carrying an ATC-inducible copy of *pgfA* on a multi-copy plasmid, cell-associated LM/LAM were extracted, separated via SDS-PAGE, and visualized by ProQ Emerald with and without ATC at the indicated timepoints.

The online version of this article includes the following source data and figure supplement(s) for figure 4:

**Source data 1.** Unprocessed lipomannan and lipoarabinomannan (LM/LAM) SDS-PAGE during PgfA depletion.

**Source data 2.** Unprocessed PIM thin-layer chromatography (TLC) during PgfA depletion.

**Source data 3.** Unprocessed plasma membrane thin-layer chromatography (TLC) during PgfA depletion.

**Source data 4.** Unprocessed, lipomannan and lipoarabinomannan (LM/LAM) SDS-PAGE, PIM, and plasma

*Figure 4 continued on next page*

*Figure 4 continued*

membrane thin-layer chromatography (TLC) during PgfA depletion, with relevant bands labeled.

**Source data 5.** Results of Tn-seq experiment comparing insertions in wild type and Δ*lamA*.

**Source data 6.** Unprocessed lipomannan and lipoarabinomannan (LM/LAM) SDS-PAGE during PgfA overexpression.

**Source data 7.** Unprocessed and labeled lipomannan and lipoarabinomannan (LM/LAM) SDS-PAGE during PgfA overexpression.

**Figure supplement 1.** Two additional biological replicates of lipomannan and lipoarabinomannan (LM/LAM) analysis during PgfA depletion.

**Figure supplement 1—source data 1.** Unprocessed lipomannan and lipoarabinomannan (LM/LAM) SDS-PAGE during PgfA overexpression.

**Figure supplement 1—source data 2.** Unprocessed PIM thin-layer chromatography (TLC) during PgfA overexpression, left-hand side.

**Figure supplement 1—source data 3.** Unprocessed PIM thin-layer chromatography (TLC) during PgfA overexpression, right-hand side.

**Figure supplement 1—source data 4.** Unprocessed plasma membrane thin-layer chromatography (TLC) during PgfA overexpression, left-hand side.

**Figure supplement 1—source data 5.** Unprocessed plasma membrane thin-layer chromatography (TLC) during PgfA overexpression, right-hand side.

**Figure supplement 1—source data 6.** Full uncropped gels with relevant bands labeled.

**Figure supplement 2.** Corynebacterial LmcA cannot functionally replace PgfA in mycobacteria.

differences in growth rate, with similar timing between successive division events of individual cells (*Figure 5—figure supplement 1*). By quantifying polar growth with an established pulse chase assay (*Aldridge et al., 2012*; *Rego et al., 2017*), we find that Δ*lamA* cells encoding an extra copy of *pgfA*, but not *mmpL3*, grow more from the old pole over the course of the cell cycle (*Figure 5A*). These data suggest that the function of PgfA or its interaction with other factors is necessary and sufficient for growth from the old pole in *M. smegmatis*.

Intriguingly, overexpression of PgfA does not lead to more growth at the new pole (*Figure 5A*). This mirrors the observation that in Δ*lamA* cells – which grow more from the new pole – neither MmpL3 nor PgfA are more abundant at this pole (*Figure 2C*). Instead, the levels of msfGFP-MurJ become increased (*Figure 2—figure supplement 3*), leading us to hypothesize that peptidoglycan synthesis increases at this site in the absence of *lamA*. To test this, we stained cells with the fluorescent D-Ala analog HADA, which is incorporated into peptidoglycan through the cytoplasmic route of synthesis (*García-Heredia et al., 2018*). In agreement with the increased MurJ abundance at the new pole, we find that Δ*lamA* cells have increased HADA staining at the new pole (*Figure 5B*). These data suggest that the molecular requirements for growth between the new and old poles may be different: with PgfA being rate limiting at the old pole and PG synthesis being rate limiting at the new pole.

## Discussion

Our understanding of the mechanisms that govern polar growth and division are at a nascent stage compared to our understanding of these processes in model rod-shaped bacteria (*Baranowski et al., 2019*; *Kieser and Rubin, 2014*). This is unfortunate because the enzymes that create and remodel the cell envelope are a rich source of anti-bacterial drug targets. In addition, the unusual mode of asymmetric polar growth – coupled with the complexity of constructing the multi-layered mycobacterial cell envelope – means that the details of growth and division in mycobacteria are almost certainly different from laterally growing rod-shaped bacteria.

Here, we investigate the function of a mycobacterial specific factor LamA. We had previously shown that LamA is, in part, responsible for asymmetric polar growth. Cells missing *lamA* grow more symmetrically and are uniformly susceptible to several drugs (*Rego et al., 2017*). How does LamA create asymmetry? We make inroads into answering this question by investigating the cellular function of MSMEG_0317, another protein of unknown function, which was predicted to interact with LamA. We show that MSMEG_0317, renamed PgfA, is involved in the trafficking of mycolic acids in

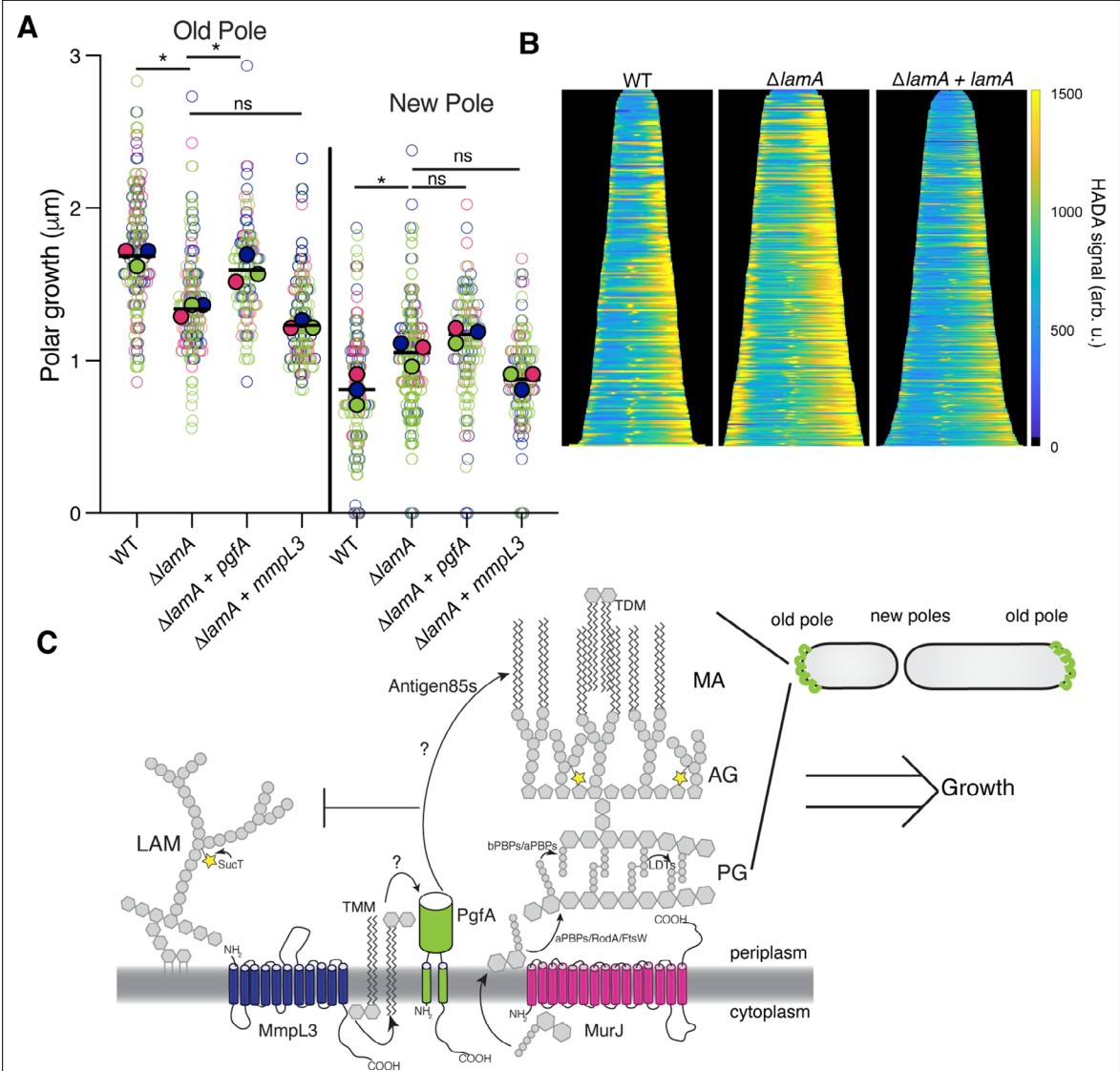

**Figure 5.** PgfA is sufficient to restore old pole growth in Δ*lamA* cells. (**A**) Cells were stained with amine-reactive dye, loaded into a microfluidic device, and imaged while perfusing dye-free media. By following the amount of unlabeled cell wall material, growth at the old and new poles was measured over time. The amount of growth incorporated per cell cycle is compared across wild type (WT) cells, Δ*lamA* cells, and Δ*lamA* cells expressing a second copy of *pgfA* or *mmpL3* driven by a strong promoter. To determine statistical significance, this procedure was repeated three times on separate days with independently grown cultures (N=3) (colors represent the replicates). For each replicate, a median was calculated (solid circles). A paired one-way ANOVA was performed comparing the medians of the replicates (aka a 'super plot'). All strains were compared against Δ*lamA*. Solid lines represent the means of the solid circles. *p<0.05; **p<0.005. (**B**) Cells were stained with a fluorescent D-alanine analog, HADA, fixed, and imaged by fluorescence microscopy. MicrobeJ was used to segment the cells and measure intensity profiles along the long axis of the cell. Profiles were then aligned by the brightest pole and ordered by length as a proxy for cell cycle. (N=194 WT; N=259 Δ*lamA*; N=335 Δ*lamA* +*lamA*) (**C**) A model for asymmetric growth. PgfA is recruited by LamA to the old pole, where, together with other proteins, it functions to organize the mycobacterial cell envelope and promote fast cell elongation at one side of the cell.

The online version of this article includes the following figure supplement(s) for figure 5:

**Figure supplement 1.** The inter-division time (the time between birth and division) was compared across strains and replicates as described in *Figure 5A*.

the periplasmic space. In fact, it is unknown how TMMs are physically placed in the mycomembrane. It has been hypothesized that MmpL3 must hand off TMMs to another protein or proteins in the periplasm before they can be incorporated into mAGP or made into TDM by antigen 85 enzymes (*Adams et al., 2021*). We find that PgfA interacts with both MmpL3 and TMM, and that cells depleted for PgfA

display phenotypes consistent with PgfA filling some intermediate role between MmpL3 and subsequent placement of mycolic acids in the mycomembrane. Further research will be needed to precisely establish PgfA's role in TMM transport; nevertheless, these results advance our understanding of how the mycomembrane is constructed.

In addition to its role in TMM transport, our data point to a more global function for PgfA in determining the composition of the cell envelope. Specifically, we find that PgfA levels negatively correlate with the abundance of LAM, and to some extent LM, two large and abundant lipoglycans in the cell envelope (*Figure 4*). These data suggest that there may be compensatory or competing mechanisms for the transport and/or synthesis of TMM and LM/LAM (*Figure 5C*). Further strengthening that argument, we find that ΔlamA cells, which have less PgfA, are more reliant on SucT – an enzyme that succinylates LAM and arabinogalactan (*Bhamidi et al., 2008*; *Palčeková et al., 2019*). It may be that TMM and LM/LAM are simply competing for space in the outer leaflet of the mycomembrane. However, to date, altered LM/LAM levels have not been reported for other TMM-trafficking mutants, suggesting that there may be more active regulation controlling the levels or transport of these two molecules. Indeed, others have speculated that succinylation itself may act to negatively regulate mycoloylation, as succinylated arabinan chains are found to be unmycoloylated (*Bhamidi et al., 2008*). Deletion of lamA results in less PgfA and MmpL3 at one side of the cell, the 'old pole', leading to less growth from that side of the cell. Increased expression of PgfA but not MmpL3 restores growth at that side of the cell, suggesting that PgfA's role in coordinating lipid trafficking, or the presence of the protein itself, is rate limiting for growth at the old pole (*Figure 5C*). Further research will be needed to understand how a TMM-binding protein influences the rate of cell wall insertion at the old pole. Indeed, insertion of new cell wall material requires the concerted effort of many synthetic enzymes, including those that polymerize and cross-link peptidoglycan. In mycobacteria, the bifunctional class A PBPs (aPBPs) are the dominant enzymes required for cell elongation. In other organisms, aPBPs are activated by inner or outer membrane-anchored proteins (*Fenton et al., 2018*; *Paradis-Bleau et al., 2010*; *Typas et al., 2010*), though no activator of PonA1 – the major aPBP in mycobacteria – has been found. Thus, it is intriguing to speculate that mycomembrane precursor transport may play a role in activating PG synthesis at the old pole in mycobacteria. What are the requirements for growth at the new pole – the pole newly formed by division? In ΔlamA, this pole grows more without an increase in PgfA or MmpL3 (*Figure 2C*); instead, enzymes involved in PG synthesis are more abundant at this site in ΔlamA. These observations suggest that the requirements for growth between the two poles are fundamentally different, at least for a certain time after the onset of pole elongation. Indeed, new poles will eventually become old poles in the next round of division. We see that PgfA and MmpL3 localize to the 'new old poles' after division, establishing asymmetry in the next generation. How does LamA affect the growth of the poles in opposite directions – inhibiting one but activating another? Future research is needed, but our data suggest that the new and old poles of mycobacteria may be distinct sites, with different molecular factors and requirements for growth. Our results suggest that LamA influences these differences to create asymmetry in individual cells, leading to a heterogeneous population better able to survive certain stresses, like antibiotics.

## Materials and methods

**Key resources table**

| Reagent type (species) or resource | Designation | Source or reference | Identifiers | Additional information |
|---|---|---|---|---|
| Strain, strain background (*Mycobacterium smegmatis*) | | | *Mycobacterium smegmatis* mc$^2$155 | Wild type *Mycobacterium smegmatis* |
| Strain, strain background (*Mycobacterium smegmatis*) | KG26 | This work | mc$^2$155 ΔMSMEG_0317 L5::pTetO-MSMEG_0317 | Parental swap strain |
| Strain, strain background (*Mycobacterium smegmatis*) | HR342 | *Rego et al., 2017* | mc$^2$155 ΔMSMEG_4265 | LamA knockout strain |
| Strain, strain background (*Mycobacterium smegmatis*) | KG97 | This work | mc$^2$155 ΔMSMEG_0317 L5::pTetO-MSMEG_0317-myc / pTetR | Strain used for the anhydrotetracyline-inducible depletion of PgfA |

*Continued on next page*

*Continued*

| Reagent type (species) or resource | Designation | Source or reference | Identifiers | Additional information |
|---|---|---|---|---|
| Strain, strain background (*Mycobacterium smegmatis*) | KG81 | This work | mc²155 ΔMSMEG_0317 pL5::pNative-MSMEG_0317 -mRFP-myc | Strain used for determining PgfA localization |
| Strain, strain background (*Mycobacterium smegmatis*) | HR388 | *Rego et al., 2017* | mc²155 L5::pG-MCK-ptb21-eGFP-wag31 tweety::ptb21-ftsZ-mCherry2B | Strain used to determine localization of FtsZ and Wag31 |
| Strain, strain background (*Mycobacterium smegmatis*) | CMG184 | This work | mc²155 tw::ptb21 mmpL3$_{TB}$-msfGFP | Strain used to determine localization of MmpL3 |
| Strain, strain background (*Mycobacterium smegmatis*) | CMG530 | This work | mc²155 ΔMSMEG_0317 L5::pTetO-MSMEG_0317-3x- FLAG tw::ptb21-mmpL3$_{TB}$-msfGFP | Strain used for co-immunoprecipitation of MmpL3 and PgfA |
| Strain, strain background (*Mycobacterium smegmatis*) | KG167 | This work | mc²155 ΔMSMEG_0317 L5::pTetO-MSMEG_0317-3x-FLAG | Control strain used for co-immunoprecipitation of MmpL3 and PgfA |
| Strain, strain background (*Mycobacterium smegmatis*) | CMG184 | This work | mc²155 tw::ptb21-mmpL3$_{TB}$-msfGFP | Strain used to determine localization of MmpL3 |
| Strain, strain background (*Mycobacterium smegmatis*) | CMG481 | This work | mc²155 ΔlamA::zeo L5:: pNative-lamA-strep tw:: ptb21-mmpL3$_{TB}$-msfGFP | Strain used to determine localization of MmpL3 in a ΔlamA background with LamA complemented under its native promoter |
| Strain, strain background (*Mycobacterium smegmatis*) | CMG557 | This work | mc²155 L5::pTetO-MSMEG_0317-myc | Strain used for measuring polar growth in a PgfA merodiploid background |
| Strain, strain background (*M. smegmatis*) | KG175 | This work | mc²155 ΔlamA L5::pTetO-MSMEG_0317-myc | Strain used for measuring polar growth in a PgfA merodiploid and LamA knockout background |
| Strain, strain background (*Mycobacterium smegmatis*) | CMG585 | This work | mc²155 L5::pTetO-mmpL3$_{TB}$-myc | Strain used for measuring polar growth in an MmpL3 merodiploid background |
| Strain, strain background (*Mycobacterium smegmatis*) | CMG587 | This work | mc²155 ΔlamA L5:: pTetO-mmpL3$_{TB}$-myc | Strain used for measuring polar growth in an MmpL3 merodiploid and LamA knockout background |
| Strain, strain background (*Mycobacterium smegmatis*) | KG154 | This work | mc²155 ΔlamA ΔMSMEG_0317 L5::pNative-MSMEG_0317 -mRFP-myc | Strain used for PgfA localization in LamA knockout background |
| Strain, strain background (*Mycobacterium smegmatis*) | KG157 | This work | mc²155 ΔlamA ΔMSMEG_0317 L5::pNative-MSMEG_0317 -mRFP-myc L5::pNative-lamA-strep | Strain used for PgfA localization in a background where lamA was complemented in LamA knockout |
| Strain, strain background (*Mycobacterium smegmatis*) | CMG206 | This work | mc²155 tw::ptb21-msfGFP-MurJ$_{TB}$ | Strain used for localization of MurJ |
| Strain (*Mycobacterium smegmatis*) | CMG199 | This work | mc²155 ΔlamA tw::ptb21-msfGFP-murJ$_{TB}$ | Strain used for localization of MurJ in a LamA knockout background |
| Strain, strain background (*Mycobacterium smegmatis*) | CMG485 | This work | mc²155 ΔlamA::zeo L5::pNative-LamA-strep tw::ptb21-msfGFP-MurJ$_{TB}$ | Strain used for localization of MurJ in a LamA knockout background where LamA was complemented under its native promoter |
| Strain, strain background (*Mycobacterium smegmatis*) | KG244 | This work | mc²155 L5::pLJR962-mmpL3-sgRNA | CRISPR-inducible MmpL3 depletion strain |
| Strain, strain background (*Mycobacterium smegmatis*) | KG289/CMG549 | This work | mc²155 L5::pLJR962- MSMEG_0317-sgRNA tw::pNative- MSMEG_0315-strep | CRISPR-inducible PgfA depletion strain |
| Strain, strain background (*Mycobacterium smegmatis*) | KG47 | This work | mc²155 ΔMSMEG_0317 L5::pTetO-MSMEG_0317-mGFPmut3-myc | C-terminal GFP tagged PgfA strain |
| Strain, strain background (*Mycobacterium smegmatis*) | KG73 | This work | mc²155 ΔMSMEG_0317 L5::pTetO-mGFP-mut3- MSMEG_0317-myc | N-terminal GFP tagged PgfA strain |
| Strain, strain background (*Mycobacterium smegmatis*) | KG167 | This work | mc²155 ΔMSMEG_0317 L5::pTetO-MSMEG_0317-3x-FLAG | Strain carrying MSMEG_0317-3×-FLAG, used for TMM pulldown |

*Continued on next page*

*Continued*

| Reagent type (species) or resource | Designation | Source or reference | Identifiers | Additional information |
|---|---|---|---|---|
| Strain, strain background (*Mycobacterium smegmatis*) | KG60 | This work | mc²155 /pTetOR- MSMEG_0317-myc | Strain carrying tetracycline inducible overexpression plasmid for PgfA |
| Strain, strain background (*Mycobacterium smegmatis*) | KG286 | This work | mc²155 Δmsmeg_0317 L5::pTetO-MSMEG_0317-ΔC | Strain carrying msmeg_0317 without c-terminal cytoplasmic tail |
| Strain, strain background (*Mycobacterium smegmatis*) | HR404 | This work | mc²155 ΔMSMEG_6398 | Antigen 85A KO strain used in TLC control experiments |
| Recombinant DNA reagent | KG147 (plasmid) | This work | DH5α / pL5-pTetO-Ncgl2760-myc | Plasmid carrying corynebacterial *ncgl2760* gene |
| Recombinant DNA reagent | KG284 (plasmid) | This work | DH5α / pL5-pTetO-Ncgl2760-MSMEG_0317-C-term | Plasmid carrying corynebacterial *ncgl2760* gene fused to cytosolic C-terminal of *msmeg_0317* gene |
| Recombinant DNA reagent | CMG541 (plasmid) | This work | Dh5a / pTweety-pNative MSMEG_0315-strep | Plasmid carrying mycobacterial gene MSMEG_0315 with a C-terminal strep tag |

## Bacterial strains and culture conditions

*M. smegmatis* mc²155 was grown in Middlebrook 7H9 broth supplemented with 0.05% Tween80, 0.2% glycerol, 5 gm/l albumin, 2 gm/l dextrose, and 0.003 gm/l catalase or plated on LB agar. *E. coli* DH5α cells were grown in LB broth or on LB agar plates. Concentrations of antibiotics used for *M. smegmatis* is as follows: 20 µg/ml zeocin, 25 µg/ml kanamycin, 50 µg/ml hygromycin, and 20 µg/ml nourseothricin. Concentrations of antibiotics used for *E. coli* used for *E. coli* are as follows: 40 µg/ml zeocin, 50 µg/ml kanamycin, 100 µg/ml hygromycin, and 40 µg/ml nourseothricin.

## Plasmid and strain construction

Strains used in this study are listed in Key resources table. Oligos and primers are listed in *Supplementary file 1*. In brief, before deleting the native copy of *MSMEG_0317 (pgfA)*, a merodiploid strain was created by inserting a second copy of *MSMEG_0317* gene under pTetO promoter using a Kan^R L5 integrating vector. Subsequently, in the merodiploid strain, an in-frame deletion was made by replacing the native copy of *MSMEG_0317 gene* with a zeocin resistance cassette flanked by loxP sites using recombineering (*van Kessel and Hatfull, 2007*). *MSMEG_0317* and its variants, including the fluorescent protein fusions, were cloned in a Nat^R L5 integrating vector for allele exchange at the L5 site. The *MSMEG_0317* depletion strain was made by transforming an episomal Hyg^R marked vector constitutively expressing TetR repressor into a strain expressing a single copy of *MSMEG_0317* controlled by the pTetO promoter. For expression of *MSMEG_0317* driven by the native promoter, 200 bp upstream of *MSMEG_0317* chromosomal locus was used. For the MmpL3 and MurJ fluorescent fusions and co-immunoprecipitation experiments, the genes were cloned from the *M. tuberculosis* genome, expressed from the ptb21 promoter and integrated in single copy at the tweety phage integration site. All plasmid constructs were made using isothermal Gibson assembly whereby insert and vector backbone shared 20–25 bp of homology.

## MSMEG_0317 depletion

### Promoter depletion

To transcriptionally deplete *MSMEG_0317* in Msm, the ATC-inducible Tet-ON system was used. The only copy of *MSMEG_0317* was driven by the pTetO promoter, while the TetR repressor was constitutively expressed in trans from an episomal vector. MSMEG_0317 was depleted by removing ATC from the medium and cells were grown for 18 hr. Subsequently, MSMEG_0317-depleted cells were re-diluted in fresh medium, also without ATC, and samples at different timepoints were taken and processed for cryo-EM. Alternatively, to avoid changes in LM/LAM abundance that have been found to be correlated with cell density and growth phase (21), at 18 hr of depletion, half of the culture was removed for lipid extraction and analysis, and replaced with fresh media. Lipid analysis was performed at timepoints before and after the decrease in OD indicating cessation of growth (*Figure 4—figure supplement 1*).

## CRISPRi depletion

To transcriptionally deplete MmpL3 and MSMEG_0317 using CRISPRi, both dCas9 and the respective guide RNAs were cloned in an L5 integrating plasmid pJR962 (*Rock et al., 2017*). The CRISPRi silencing of *mmpL3/MSMEG_0317* was induced by adding ATC to a final concentration of 100 ng/μl. The induction was carried out for the indicated timepoints.

## Time-lapse and fluorescence microscopy

An inverted Nikon Ti-E microscope was used for the time-lapse and snapshot imaging. An environmental chamber (Okolabs) maintained the sample at 37°C. To reduce phototoxicity exposure times were kept below 100 ms for excitation with 470 nm and 300 ms for excitation with 550 nm.

### Time lapse

Exponentially growing cells were cultivated in an B04 microfluidics plate from CellAsic, continuously supplied with fresh 7H9 medium, and imaged every 15 min using a 60× 1.4 NA Plan Apochromat phase contrast objective (Nikon). Fluorescence was excited using the Spectra X Light Engine (Lumencor), separated using single- or multi-wavelength dichroic mirrors, filtered through single bandpass emission filters, and detected with an sCMOS camera (ORCA Flash 4.0). Filters are: GFP (Ex: 470/24; Em: 515/30 or 525/50); mRFP (Ex: 550/15; Em: 595/44 or 630/75).

### RADA pulse chase

To generate the data shown in *Figure 2—figure supplement 1*, cells from a log-phase culture were stained overnight with the fluorescent D-amino acid dye RADA at a final concentration of 2 mM. The next morning, those cultures were split into two tubes, one of which received 100 ng anhydrous tetracycline (aTC) to deplete either MmpL3 or PgfA; the other tube did not receive aTC as a control. After growing for 5 hr of depletion, all cells were washed of fluorescent dye by centrifugation and resuspension in growth media, and allowed to outgrow another 3 hr. All samples were then imaged under 1% agarose Hartmans-de Bont (HdB) pads. RADA dye was captured by using a custom TRITC filter system (Ex: 560/40 or 550/10; Em: 630/73) with an excitation time of 50 ms.

## Kymograph analysis and image analysis

Time-lapse images were analyzed in open-source image analysis software Fiji (30) and a custom MATLAB program was used to generate kymographs. Source code will be uploaded to GitHub. Specifically, for a single cell, in Fiji, a 5-pixel wide segmented line was drawn from the new pole to the old pole at each timepoint during the cell cycle. This was repeated on 20–50 cells. These line profiles were then imported into MATLAB, where a custom script was used to generate an average kymograph by 2D interpolation of the individual kymographs. For the demographs shown in *Figure 5*, cells were segmented and intensity profiles measured using MicrobeJ. Output intensity profiles were then transferred to MATLAB, sorted by cell length, and aligned by the brightest pole for visualization. For pulse chase assays shown in *Figure 5*, cells were measured using a 7-pixel wide segmented line drawn from the new pole to the old pole at two points during the cell cycle: birth and the frame before division. These line profiles were then imported into MATLAB, where a custom script was used to measure the amount of growth at each pole during a single generation.

## N-x-AlkTMM-C15-mediated protein capture

### Preparation of cell lysate

*M. smegmatis* expressing 3× FLAG-tagged MSMEG_0317 starter culture was generated by inoculating a single colony from a freshly streaked LB agar plate supplemented with zeocin and nourseothricin (20 μg/ml each) into 10 ml liquid 7H9 medium in a sterile culture tube. The starter culture was grown until mid-logarithmic phase and then diluted to OD600 0.2 with 7H9 medium to a final volume of 80 ml. The culture was split into two equal volumes in 125 ml sterilized culture flasks. To one flask, N-x-AlkTMM-C15 was added to a final concentration of 100 μM with (final DMSO concentration of 2%), while the other flask was left untreated as a DMSO control (final DMSO concentration of 2%). Both flasks were incubated with shaking until OD600 0.8 was reached. The cells were pelleted by centrifugation at 6500× *g* at 4°C for 10 min. The cell pellets were washed twice with PBS, resuspended

in 6 ml PBS, and split into two equal volumes. One aliquot was exposed to UV irradiation for 30 min with a 5 W 365 nm UV bench lamp (UVP) while the other was left unexposed as a control. The cell pellets were collected by centrifugation and resuspended in 600 µl lysis buffer (2 mg/ml lysozyme, 1 mM phenylmethylsulfonyl fluoride in 1× PBS), transferred to scintillation vials, and incubated at 37°C for 2 hr. The mixtures were transferred to 1.5 ml screw-cap vials containing 0.25 ml of 0.1 mm zirconia/silica beads (BioSpec Products) and subjected to bead beating 3× for 1 min each using a FastPrep-24 bead beater (MP Biomedicals). The lysate was transferred to a scintillation vial, then SDS was added to a final concentration of 2% from a 20% SDS stock. Cell extracts were incubated at 60°C for 2 hr with constant stirring. The lysates were transferred to microcentrifuge tubes and centrifuged at 3200× $g$ for 10 min at 4°C. The supernatant was collected and stored in separate tubes at 4°C until use.

## CuAAC and affinity enrichment

To reduce SDS concentration, 500 µl of methanol/chloroform (2:1 v/v) was added to 184 µl of cell lysate. The resulting protein precipitate was centrifuged at 18,000× $g$ at 4°C for 10 min and the supernatant was discarded. The precipitate was air-dried and solubilized using 184 µl of 0.5% SDS/0.05% LDAO buffer. Copper-catalyzed azide-alkyne cycloaddition (CuAAC) was carried out by sequential addition of 1 mM AzTB (azido-TAMRA-biotin (4 µl), Click Chemistry Tools), 60 mM sodium ascorbate (4 µl), 6.4 mM TBTA ligand (4 µl), and 50 mM copper sulfate (4 µl) to give a final volume of 200 µl. The final reagent concentration was 20 µM AzTB; 1.2 mM sodium ascorbate; 128 µM TBTA; and 1 mM copper(II) sulfate. The mixture was thoroughly mixed by pipetting up and down and the reaction was incubated for 2 hr in the dark with constant agitation at 37°C. Excess AzTB was removed by precipitating proteins in methanol/chloroform (2:1 v/v), discarding supernatant, and solubilizing protein pellets in 200 µl 0.5% SDS/0.05% LDAO buffer as described above. Protein concentration was determined by Bradford assay. Fifteen µl of each sample was saved as input sample. The remaining sample was mixed with 40 µl Pierce avidin-agarose beads (Thermo Fisher Scientific) that had been pre-washed 3× with 0.5% SDS/0.05% LDAO buffer (100 µl). The bead mixture was incubated at room temperature with constant rotation for 2 hr. The beads were washed 3× with 0.5% SDS/0.05% LDAO buffer (100 µl) followed by 3× with PBS (100 µl) with centrifugation at 1000× $g$ for 1 min between each wash. Bound proteins were eluted by boiling at 95°C for 15 min in 30 µl of 4× sample buffer.

## SDS-PAGE and Western blot

Five µg of input and 10 µl of output was resolved by SDS-PAGE by 4–20% acrylamide gels and analyzed by in-gel fluorescence using a Typhoon FLA 7000 (GE Healthcare Life Science) using the rhodamine channel to detect TAMRA-labeled proteins.

The gel was fixed for 15 min (40% ethanol and 10% acetic acid in DI water), rinsed 3× with DI water 10 min each and stained overnight with agitation in QC colloidal Coomassie stain (Bio-Rad). The gel was rinsed with DI water 3× for 10 min and imaged using a ChemiDoc Touch Imaging System (Bio-Rad) and processed by Image Lab software (Bio-Rad).

The above conditions were used to generate samples for Western blot analysis. Five µg input controls and 10 µl eluted proteins were resolved by SDS-PAGE. After gel electrophoresis, proteins were transferred onto an Immun-Blot PVDF membrane (Bio-Rad). The PVDF membrane was equilibrated in ethanol and blotting filter paper (Thermo Fisher Scientific) was equilibrated in transfer buffer (25 mM Tris, 193 mM glycine, and 20% ethanol) before placement in a transfer cassette. Proteins were transferred electrophoretically at a constant voltage of 25 V for 7 min using a Trans-Blot Turbo Transfer System (Bio-Rad). After transfer, the membrane was blocked overnight at 4°C in 5% dry milk in Tris-buffered saline containing Tween 20, pH 7.6 (50 mM Tris, 0.5 M NaCl, 0.02% Tween 20 [TBST]). Anti-FLAG-HRP (Thermo Fisher Scientific) was used at 1:1000 dilution using 2% dry milk in TBST. The membrane was incubated with antibody at 4°C overnight with constant rocking. The membrane was washed 3× for 10 min each with TBST. The membrane was treated with SuperSignal West Pico PLUS chemiluminescent reagent (Thermo Fisher Scientific) and chemiluminescence was detected on a ChemiDoc Touch Imaging system (Bio-Rad).

## Tn-seq

Transposon libraries in wild type and Δ*lamA* (HR342) were prepared as and sequenced described elsewhere (*Dragset et al., 2019*). This was done independently on two separate days for a total of

two biological replicates in each strain. The TRANSIT and TPP python packages were used to map insertions to the mc²155 genome and quantitatively compare insertions across stains, using the 'resampling' option (*DeJesus et al., 2015*).

## Co-immunoprecipitation

### Sample preparation and immunoprecipitation

Mycobacterial cultures were grown to mid-log phase as described above. Protocol for cross-linking was adapted from *Belardinelli et al., 2019*. Cells were washed with 1× phosphate buffered saline (PBS) once. Pellets were resuspended in 1 ml of 1× PBS with 1.25 mM dithiobis(succinimidyl propionate) and incubated for 30 min at 37°C for cross-linking. After incubation, cells were pelleted at 10,000× *g* for 5 min at room temperature and the supernatant was discarded. The pellet was resuspended in lysis buffer (50 mM Tris-HCl, pH 7.4; 150 mM NaCl; 10 ug/ml DNase I; one tablet Roche cOmplete EDTA free protease inhibitor cocktail; and 0.5% Igepal Nonidet P40 Substitute) and lysed with a BeadBug Microtube Homogenizer at 4000 rpm six times for 30 s each, icing in between. Lysed cells were spun down at 15,000× *g* for 15 min at 4°C, and the supernatant was transferred to a clean Eppendorf tube. Lysates were incubated, where indicated, with either GFP-Trap Magnetic Agarose (Chromotek) or magnetic a-FLAG M2 beads (Sigma-Aldrich) and incubated at 4°C overnight, rotating. After incubation, samples were spun down at 2500× *g* for 1 min at room temperature and flow through was discarded. Beads were washed three times with non-detergent wash buffer (10 mM Tris-HCl, pH 7.4; 150 mM NaCl, 0.5 mM EDTA). GFP-Trap samples were eluted with 2× Laemmli Buffer (Bio-Rad) prepared with 50 mM (DTT) and boiled at 95°C for 5 min. FLAG M2 beads were eluted twice with 3× FLAG peptide (Sigma-Aldrich) for 30 min rotating at 4°C. All samples not already treated with Laemmli Buffer + DTT were prepared for Western blotting by addition of Laemmli Buffer + DTT and boiled at 95°C for 5 min, to reverse all cross-links.

### Western blot verification

Samples were run on NuPAGE (Thermo Fisher) 4–12% Bis-Tris gels in 1× MOPS-SDS running buffer. Proteins were transferred to nitrocellulose membranes and probed with 1:1000 a-FLAG primary (Sigma-Aldrich, clone M2) and 1:5000 a-mouse secondary (Thermo Fisher, Superclonal A28177). SuperSignal West Femto Extended Duration Substrate (Thermo Fisher) was used for chemiluminescent visualization on an Amersham ImageQuant 800 system.

## Lipid extraction and analysis

### TMM/TDM

Crude lipids were extracted from equal wet cell pellet weights of the either *mmpL3* depletion or *MSMEG_0317*-depleted cells after 9, 12, and 15 hr with or without ATC with 2:1 chloroform/methanol mixture for 12 hr. The organic layer was separated from the cell debris centrifugation at 4000× *g*. The organic extract was air-dried overnight. The dried extract was dissolved in 50 μl of 2:1 chloroform/methanol. TMMs and TDMs were separated on by high-performance thin layer chromatography (HPTLC) (Silica gel 60, EMD Merck) using chloroform/methanol/water (25:4:9:0.4). TMMs/TDMs were visualized by spraying the TLC sheet with 20% 1-napthaol in 5% sulfuric acid and charring the plate at 110°C Trehalose containing lipids (TMMs/TDMs) appear as purple bands after charring.

### LM/LAM

Extraction, purification, and analysis of lipids were as described previously (*Rahlwes et al., 2019*). Briefly, crude lipids were extracted from equal wet cell pellet weights of the MSMEG_0317 depletion strains after 18, 24, 27, 30, and 33 hr with or without ATC. After lipid extraction using chloroform/methanol, LM and LAM were extracted from the delipidated pellet by incubation with phenol/water (1:1) for 2 hr at 55°C. Phospholipids and PIMs extracted by chloroform/methanol were further purified by *n*-butanol/water phase partitioning, and separated by HPTLC (Silica gel 60, EMD Merck) using chloroform/methanol/13 M ammonia/1 M ammonium acetate/water (180:140:9:9:23). Phospholipids were visualized via cupric acetate staining. PIMs were visualized with orcinol staining as described (*Sena et al., 2010*). LM/LAM samples were separated by SDS-PAGE (15% gel) and visualized using ProQ Emerald 488 glycan staining kit (Life Technologies). To detect LM/LAM in culture supernatants,

the supernatants were initially treated with a final concentration of 50 µg/ml Proteinase K for 4 hr at 50°C. The treated supernatants were electrophoresed on 15% SDS-PAGE. LM/LAM were blotted onto nitrocellulose at 20 V for 45 min using semi-dry transfer method. Post transfer, the membrane was blocked by 5% milk in Tris-buffered saline supplemented with 0.05% Tween-20 (TBST). The blocked membrane was then probed overnight with CS-35 antibody (BEI Resources, NIH) at 1:250 dilution at 4°C. The membrane was washed with TBST five times for 5 min each. Post washing, it was probed with anti-mouse secondary for 1 hr at room temperature. Membrane was then washed five times for 5 min with TBST. Thermo Scientific's west dura chemiluminescent reagent was used to develop the membrane.

## Sample preparation and image collection for cryo-EM

Wild type and MSMEG_0317 depleted *M. smegmatis* cells were pelleted, washed twice with 1× PBS, and suspended in ~20 µl PBS. The culture was subsequently deposited onto freshly glow-discharged holey carbon grids. The grids were then blotted with filter paper manually for about 4 s and rapidly frozen in liquid ethane. The frozen grids were transferred into a 300 kV Titan Krios electron microscope (Thermo Fisher Scientific) equipped with a K2 Summit direct detector and a quantum energy filter (Gatan, Pleasanton, CA). Cryo-EM movie stacks were collected using SerialEM (*Mastronarde, 2005*). MotionCor2 (*Zheng et al., 2017*) was used for drift correction of the cryo-EM movie stacks. The gray levels of each micrograph are obtained using MATLAB.

## Experimental replicates

Biological replicates are defined as independent cultures grown in parallel or on separate days. Technical replicates are defined at the same culture, measured independently. All the experiments were performed at least twice – often three or more times – with biological replicates.

## Acknowledgements

We thank members of the Rego lab for helpful suggestions. This work was supported by the National Institutes of Health, R01AI148255 to EHR, who was also supported by the Searle and Pew Foundations. BMS was supported by the National Science Foundation (CAREER Award 1654408). CW and JL were supported by the NIH grants R01AI087946 and R01GM110243.

## Additional information

### Funding

| Funder | Grant reference number | Author |
| --- | --- | --- |
| National Institute of Allergy and Infectious Diseases | R01AI148255 | E Hesper Rego |
| National Science Foundation | 1654408 | Benjamin M Swarts |
| National Institute of Allergy and Infectious Diseases | R01AI087946 | Jun Liu |
| National Institute of General Medical Sciences | R01GM110243 | Jun Liu |
| Pew Charitable Trusts | | E Hesper Rego |
| Searle Scholars Program | | E Hesper Rego |

The funders had no role in study design, data collection and interpretation, or the decision to submit the work for publication.

### Author contributions

Kuldeepkumar R Gupta, Conceptualization, Data curation, Formal analysis, Investigation, Methodology, Writing - original draft, Writing - review and editing; Celena M Gwin, Conceptualization, Data

curation, Formal analysis, Investigation, Writing - review and editing; Kathryn C Rahlwes, Chunyan Wang, Resources, Formal analysis, Investigation, Writing - review and editing; Kyle J Biegas, Resources, Formal analysis, Investigation, Methodology, Writing - review and editing; Jin Ho Park, Investigation, Writing - review and editing; Jun Liu, Resources, Formal analysis, Supervision, Funding acquisition, Writing - review and editing; Benjamin M Swarts, Resources, Formal analysis, Supervision, Funding acquisition, Writing - original draft, Writing - review and editing; Yasu S Morita, Conceptualization, Formal analysis, Supervision, Funding acquisition, Writing - original draft, Writing - review and editing; E Hesper Rego, Conceptualization, Data curation, Formal analysis, Supervision, Funding acquisition, Investigation, Writing - original draft, Writing - review and editing

### Author ORCIDs
Kuldeepkumar R Gupta  http://orcid.org/0000-0001-7625-4224
Celena M Gwin  http://orcid.org/0000-0002-8928-7970
Jun Liu  http://orcid.org/0000-0003-3108-6735
E Hesper Rego  http://orcid.org/0000-0002-2973-8354

### Decision letter and Author response
Decision letter https://doi.org/10.7554/eLife.80395.sa1
Author response https://doi.org/10.7554/eLife.80395.sa2

## Additional files

### Supplementary files
• Supplementary file 1. A list of primers and oligos used in this study.

• MDAR checklist

### Data availability
All data generated or analyzed during this study are included in the manuscript and supporting files. Source Data files have been provided for blots, gels and TLC figures, and the Tn-seq data shown in Figure 4.

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
