## [Editor Report]

This manuscript tackles the important and fundamental question of what proteins regulate asymmetrical cell division in Mycobacteria. This strong study will be of interest to all individuals interested in bacterial physiology and the conclusions are well-supported by the data.

---

## [Decision Letter]

**Decision letter after peer review:**

Thank you for submitting your article, "An essential periplasmic protein coordinates lipid trafficking and is required for asymmetric polar growth in mycobacteria" for consideration by *eLife*. Your article has been reviewed by 3 peer reviewers, one of whom is a member of our Board of Reviewing Editors, and the evaluation has been overseen by Suzanne Pfeffer as the Senior Editor. The reviewers have opted to remain anonymous.

Essential revisions:

Please see the specific comments from the reviewers below. To summarize:

1) The TLC data are qualitative and difficult to assess significance. Quantitative data from several experiments should be included.

2) The conclusions for whether LmcA and PgfA are (or are not) orthologs are not strongly supported and should be revisited.

3) Overall, please re-assess the specific conclusions of the manuscript and either provide data to support the conclusions or temper claims. Details follow below.

*Reviewer #1 (Recommendations for the authors):*

– Conclusions in Lines 100-101: The inability of LmcA to substitute for PgfA does not mean these proteins are not orthologs, there are many other factors that contribute to heterologous complementation, including the similarity of other binding partners or chaperones. The data that may support the proteins are not orthologs is the absence of an effect on LAM in the PgfA depletion mutant. However, the overexpression of PgfA does lead to changes in LAM. So both proteins still have connections to these pathways and it is not convincing that the proteins are not orthologs performing functions specific to the physiology of the different bacteria.

– In lines 113-116, the authors reference the data in Figure 1B as being the differences in colonies that grew while performing the allele swapping strategy with the empty vector. Did the colonies that grew actually exhibit a loss of PfgA? Or did these isolates still contain the WT gene? It seems important to understand this in order to make conclusions about essentiality.

– Conclusions in Lines 150-152– how do the experiments with the RFP tagged PgfA show that PgfA is required for polar growth? The experiments in 1C support that PgfA is involved in cell elongation, but most strikingly it appears to be involved through the envelope integrity. But the polarity of growth is never measured during PgfA depletion. Then the experiments with the RFP tagged PgfA only show localization, but do not test what processes PgfA is required for. In this regard, line 347 is also not supported by the data it references. The authors never actually show that the pgfA mutant is deficient in polar growth.

– The authors first mention in line 199 that pgfA is in an operon. Did the authors check if there were polar effects in the mutant used in Figures 1 and 2A? For the experiments in Figure 2D, was the expression of MSMEG_0315 at WT levels at the heterologous site? What is MSMEG_0315? Is there a conserved homolog in Mtb?

– For lines 215-216, the authors claim their data suggest that cells depleted for PgfA accumulate TMM, but this is not shown, much less whether it accumulates to toxic levels. Later in the manuscript, the authors perform the TLCs, so this conclusion should follow the experimental data to support it. In addition, some of the differences in the TLCs are rather small. Replicate TLCs could be quantified, graphed, and stats performed to determine the rigor of the data.

– Do the authors have an interpretation of why TMM also increases in the overexpression strain in Figure 3D, and how does this fit into their model?

– Lines 259-260 – it could be more direct to reword the sentence so that it is clear that it is unknown what region of PgfA binds the TMM analog.

– Is there a reason that in the section starting on line 281, the authors call the corynebacterial homolog NCgI2760 instead of LmcA? Are these the same protein?

*Reviewer #2 (Recommendations for the authors):*

1. The TLC data in Figures 3C and D are qualitative and difficult to assess significance. Quantitative data on the relative abundance of the TDM and TMM bands from several experiments should be included.

2. The Results section entitled "The function of PgfA is specific to mycobacteria and its abundance negatively correlates with the abundance of lipoglycans LM/LAM" is quite long and spends a lot of time discussing data indicating that PgfA is not an ortholog of NCgl2760 from C. glutamicum. While this is useful information, it detracts from the main thrust of the manuscript-the contribution of PgfA to asymmetric growth. I recommend revising this section to reduce its length and increase focus.

*Reviewer #3 (Recommendations for the authors):*

1) The authors propose that PgfA is involved in the traffic of mycolic acids in the periplasmic space. I think the data support this idea. However, I think it would strengthen the paper to discuss in more detail what steps in the trafficking process remain unknown and where PgfA might fit into it. For example, is it potentially involved in extracting TMM from the membrane and handing it off to Ag85 proteins for transport? Are there other possibilities? More specific context such as this would improve the Discussion section and make it more impactful.

2) I am not convinced that the depletion of PgfA appreciably changes the levels of TMM or TDM in cells. Firstly, it would be helpful to know how the authors assigned the TLC spots to TMM and TDM? Controls would be helpful here, such as the inhibition of TMM synthesis to show which spots disappear. Also, in addition to adding better controls to assign the spots, some quantitation is needed. I don't see much in the way of changes to the TMM and TDM spots by eye when PgfA is depleted. In addition, there are new spots appearing at the top of the TLC plate in the PgfA-depleted cells. What species are these? Their identification could shed more light on PgfA function as they are unique to PgfA depletion and not observed in the MmpL3 depletion. At a minimum, this observation should be highlighted in the text given how striking it is (much more striking than the TMM/TDM changes).

3) The topology experiments are not especially convincing. The reduced fluorescence of the N-terminal fusion makes it hard to draw strong conclusions from this construct. Is there a small amount of protein with GFP inside that is non-functional combined with a larger fraction of non-fluorescent protein with GFP outside that accounts for the functionality of the construct? I think it is probably sufficient to rely on the topology prediction supported by the C-in result from the C-terminal GFP fusion. These combine to make a reasonable case for the proposed topology as long as the caveat is made that further experiments are needed to definitively establish the topology.

4) As mentioned above, I do not find the argument that NCgl2760 has a different function than PgfA to be convincing. This section should be (significantly) revised to account for the much more reasonable conclusion that cross complementation does not work simply because of divergent protein-protein interactions and/or mycolic acid length between organisms. The fact that PgfA/ NCgl2760 has differential effects on LM/LAM accumulation in their respective host organisms could simply be due to differential stress/regulatory responses downstream of the same primary defect.

5) The overexpression studies in Figure 5 have several problems. Firstly, were the results normalized for growth rate changes? It seems like the growth rate could significantly affect the outcome of the experiment and the relative elongation at the different poles observed. Secondly, it is concluded that PfgA overproduction but not MmpL3 overproduction restores asymmetric growth to δ-lamA cells. The results for these two strains don't look all that different to me. Some statistics are needed here.

6) Line 389: "regulating" is misused here and in other sections of the paper. Just because the inactivation of PgfA causes a change in envelope structure/composition does not mean it is a regulator. The regulator designation implies that it is responsive to stimuli, but there is no data supporting this designation. It seems more likely to me that PgfA is a core component of the transport apparatus and that its inactivation alters the envelope because the normal process is disrupted, not because it is a regulator.

7) Lines 402-415: I think some speculation is important to include in a Discussion section. However, the connection for PgfA with peptidoglycan synthesis is a bit too much of a leap for this paper. I would remove this paragraph.

8) Line 418-420: Rather than different growth requirements at the two poles, couldn't the results be interpreted as a competition between the two poles for common precursors/protein components? A competition model involving some positive feedback at the favored pole seems simpler and therefore more likely.

9) Line 215-219: Reference to TMM accumulation is made, but this comes well before the TMM levels are quantified later in the paper and the figure is not referenced. Some rewriting is needed to improve the flow of the paper here. Also, as mentioned above, I do not think the data showing TMM accumulation is convincing as presented.

10) Figure 2B: Please show the blots for the input material as well as the blots for anti-GFP as well as anti-FLAG blots.

11) Figure 2C: Make the font bigger for PgfA and MmpL3 or find some other way to make the labels clearer.

---

## [Author Response]

Essential revisions:Please see the specific comments from the reviewers below. To summarize:1) The TLC data are qualitative and difficult to assess significance. Quantitative data from several experiments should be included.2) The conclusions for whether LmcA and PgfA are (or are not) orthologs are not strongly supported and should be revisited.3) Overall, please re-assess the specific conclusions of the manuscript and either provide data to support the conclusions or temper claims. Details follow below.

We thank the reviewers for their time and thoughtful commentary. To address their concerns, we have quantified the TLC data, and determined statistical significance. These data support our conclusions and can be found in Figure 3E-F. We have also revised the text to de-emphasize our studies and interpretations around LmcA and PgfA complementation.

Reviewer #1 (Recommendations for the authors):– Conclusions in Lines 100-101: The inability of LmcA to substitute for PgfA does not mean these proteins are not orthologs, there are many other factors that contribute to heterologous complementation, including the similarity of other binding partners or chaperones. The data that may support the proteins are not orthologs is the absence of an effect on LAM in the PgfA depletion mutant. However, the overexpression of PgfA does lead to changes in LAM. So both proteins still have connections to these pathways and it is not convincing that the proteins are not orthologs performing functions specific to the physiology of the different bacteria.

In response to comments from all authors, we have we-written this section and de-emphasized the notion that PgfA and LmcA are not orthologs. Please see lines 313-335.

– In lines 113-116, the authors reference the data in Figure 1B as being the differences in colonies that grew while performing the allele swapping strategy with the empty vector. Did the colonies that grew actually exhibit a loss of PfgA? Or did these isolates still contain the WT gene? It seems important to understand this in order to make conclusions about essentiality.

The few colonies that grew when exchanged for an empty vector control were verified to be double integrants, meaning that they contain a copy of *pgfA* as well as the vector control at the L5 site. We think that these data, in addition to the data found in Figure 1C and Figure 1 —figure supplement 1A,B, make a strong case for *pgfA* being an essential gene.

– Conclusions in Lines 150-152– how do the experiments with the RFP tagged PgfA show that PgfA is required for polar growth? The experiments in 1C support that PgfA is involved in cell elongation, but most strikingly it appears to be involved through the envelope integrity. But the polarity of growth is never measured during PgfA depletion. Then the experiments with the RFP tagged PgfA only show localization, but do not test what processes PgfA is required for. In this regard, line 347 is also not supported by the data it references. The authors never actually show that the pgfA mutant is deficient in polar growth.

To verify that PgfA and MmpL3 are important for polar growth, we have performed a new experiment to visualize the amount of new growth added at the poles in cells depleted for these proteins. These data can be found in Figure 1 —figure supplement 2, and in text on lines 128-132 and 218.

– The authors first mention in line 199 that pgfA is in an operon. Did the authors check if there were polar effects in the mutant used in Figures 1 and 2A? For the experiments in Figure 2D, was the expression of MSMEG_0315 at WT levels at the heterologous site? What is MSMEG_0315? Is there a conserved homolog in Mtb?

MSMEG_0315 is predicted to be a transmembrane protein of unknown function that is conserved in Mtb, Rv0226c. Tn-seq and CRISPRi studies predict these genes to be essential. There have not been any studies on this protein in mycobacteria. However, there is one report where it is shown to be involved in the transport of cornomycolates in *C. glutamicum* (Cashmore *et al. Biomolecules* 2021).

The mutant used in Figures 1 and 2A was generated by an in-frame deletion of *msmeg_0317.* As we were able to make this mutant, in a background with only *pgfA* at phage site, we inferred that this did not interrupt the expression of the second gene of the operon, *msmeg_0315*, as it is predicted to be essential. Additionally, the ATC-inducible *pgfA* was at a phage site, uncoupling it from the expression of *msmeg_0315*, and so we feel confident that the phenotypes we report are due to changes in MSMEG_0317 (PgfA) levels. However, we were concerned about the known polar effects of CRISPRi, and we have verified that MSMEG_0315 is being expressed in this strain background by Western blot Figure 2 —figure supplement 4.

– For lines 215-216, the authors claim their data suggest that cells depleted for PgfA accumulate TMM, but this is not shown, much less whether it accumulates to toxic levels. Later in the manuscript, the authors perform the TLCs, so this conclusion should follow the experimental data to support it. In addition, some of the differences in the TLCs are rather small. Replicate TLCs could be quantified, graphed, and stats performed to determine the rigor of the data.

We have removed the conclusion of line 215-216 in the revised manuscript and put it in the appropriate section. See Figure 3E-F for quantification of TLC.

– Do the authors have an interpretation of why TMM also increases in the overexpression strain in Figure 3D, and how does this fit into their model?

We agree that there appears to be a small increase in TMM in the PgfA overexpression, but this difference is not statistically significant across replicates (Figure 3E-F).

– Lines 259-260 – it could be more direct to reword the sentence so that it is clear that it is unknown what region of PgfA binds the TMM analog.

Reworded on lines 270.

– Is there a reason that in the section starting on line 281, the authors call the corynebacterial homolog NCgI2760 instead of LmcA? Are these the same protein?

We have updated the manuscript to refer to NCgl2760 as LmcA throughout to avoid confusion.

Reviewer #2 (Recommendations for the authors):1. The TLC data in Figures 3C and D are qualitative and difficult to assess significance. Quantitative data on the relative abundance of the TDM and TMM bands from several experiments should be included.

See above.

2. The Results section entitled "The function of PgfA is specific to mycobacteria and its abundance negatively correlates with the abundance of lipoglycans LM/LAM" is quite long and spends a lot of time discussing data indicating that PgfA is not an ortholog of NCgl2760 from C. glutamicum. While this is useful information, it detracts from the main thrust of the manuscript-the contribution of PgfA to asymmetric growth. I recommend revising this section to reduce its length and increase focus.

We thank the reviewer for this insightful comment that will improve the flow and comprehensibility of our manuscript to the reader. The text has been edited to increase clarity and de-emphasize this section.

Reviewer #3 (Recommendations for the authors):1) The authors propose that PgfA is involved in the traffic of mycolic acids in the periplasmic space. I think the data support this idea. However, I think it would strengthen the paper to discuss in more detail what steps in the trafficking process remain unknown and where PgfA might fit into it. For example, is it potentially involved in extracting TMM from the membrane and handing it off to Ag85 proteins for transport? Are there other possibilities? More specific context such as this would improve the Discussion section and make it more impactful.

We thank the reviewer for this suggestion. We have reorganized the discussion and have included a paragraph (lines 427-436) in the discussion on this important topic.

2) I am not convinced that the depletion of PgfA appreciably changes the levels of TMM or TDM in cells. Firstly, it would be helpful to know how the authors assigned the TLC spots to TMM and TDM? Controls would be helpful here, such as the inhibition of TMM synthesis to show which spots disappear. Also, in addition to adding better controls to assign the spots, some quantitation is needed. I don't see much in the way of changes to the TMM and TDM spots by eye when PgfA is depleted. In addition, there are new spots appearing at the top of the TLC plate in the PgfA-depleted cells. What species are these? Their identification could shed more light on PgfA function as they are unique to PgfA depletion and not observed in the MmpL3 depletion. At a minimum, this observation should be highlighted in the text given how striking it is (much more striking than the TMM/TDM changes).

We agree that this is an important point. TMM and TDM spots were assigned by several methods. First, TLCs were performed using established protocols to get a general sense of how the lipids migrated. Secondly, we depleted or deleted genes known to affect TMM and TDM levels. We show MmpL3 depleted cells in the main text, but now include TLC data from an antigen-85a KO, as well. Lastly, as the reviewer suggested, we treated cells with isoniazid to inhibit TMM synthesis and see the spots corresponding to TMM and TDM decrease. These data can be found in Figure 3 —figure supplement 3. Together, these controls were used to appropriately assign spots on the TLCs. Of note, we also obtained purified TMM and TDM controls from *Mycobacterium tuberculosis* (ATCC NR-48784 and NR-14844) to use as controls. However, species differences meant that these lipids exhibit significant differences in migration patterns and therefore were not appropriate controls for our experiments.

The new spots appearing on the top of TLC plates in PgfA-depleted cells are glycopeptidolipids (GPLs). These outer membrane lipids are responsible colony morphology, biofilm formation and sliding motility of Msm and other NTM species. We think that the mycomembrane disruption caused due to PgfA depletion might contribute to better extraction of GPLs from the depleted cells. However, we have not observed consistent patterns of these bands across replicates in the different genetic backgrounds, and do not feel comfortable drawing conclusions about their significance.

3) The topology experiments are not especially convincing. The reduced fluorescence of the N-terminal fusion makes it hard to draw strong conclusions from this construct. Is there a small amount of protein with GFP inside that is non-functional combined with a larger fraction of non-fluorescent protein with GFP outside that accounts for the functionality of the construct? I think it is probably sufficient to rely on the topology prediction supported by the C-in result from the C-terminal GFP fusion. These combine to make a reasonable case for the proposed topology as long as the caveat is made that further experiments are needed to definitively establish the topology.

We have revised this section (lines 261-266).

4) As mentioned above, I do not find the argument that NCgl2760 has a different function than PgfA to be convincing. This section should be (significantly) revised to account for the much more reasonable conclusion that cross complementation does not work simply because of divergent protein-protein interactions and/or mycolic acid length between organisms. The fact that PgfA/ NCgl2760 has differential effects on LM/LAM accumulation in their respective host organisms could simply be due to differential stress/regulatory responses downstream of the same primary defect.

We agree and have re-written this section to deemphasize the complementation studies.

5) The overexpression studies in Figure 5 have several problems. Firstly, were the results normalized for growth rate changes? It seems like the growth rate could significantly affect the outcome of the experiment and the relative elongation at the different poles observed. Secondly, it is concluded that PfgA overproduction but not MmpL3 overproduction restores asymmetric growth to δ-lamA cells. The results for these two strains don't look all that different to me. Some statistics are needed here.

As demonstrated in Figure 5 —figure supplement 1, there is no difference in growth rate between the strains used in Figure 5A. We now show replicate data in Figure 5A. We determined statistical significance by comparing the averages of the replicate single-cell distributions (aka a “super plot”). This much more rigorous analysis confirms our original conclusion that increased expression of PgfA restores growth at the old pole in Δ*lamA* cells.

6) Line 389: "regulating" is misused here and in other sections of the paper. Just because the inactivation of PgfA causes a change in envelope structure/composition does not mean it is a regulator. The regulator designation implies that it is responsive to stimuli, but there is no data supporting this designation. It seems more likely to me that PgfA is a core component of the transport apparatus and that its inactivation alters the envelope because the normal process is disrupted, not because it is a regulator.

We agree and have made the appropriate changes to remove “regulating” from the manuscript in reference to PgfA.

7) Lines 402-415: I think some speculation is important to include in a Discussion section. However, the connection for PgfA with peptidoglycan synthesis is a bit too much of a leap for this paper. I would remove this paragraph.

We have softened some of the language in this section as to not over-reach the conclusions of our data. However, we believe that this is an interesting avenue of research given the trend of peptidoglycan synthesis activation by lipoproteins in other bacterial species. It follows that the pathways that govern lipid trafficking may be involved in similar regulatory mechanisms in mycobacteria.

8) Line 418-420: Rather than different growth requirements at the two poles, couldn't the results be interpreted as a competition between the two poles for common precursors/protein components? A competition model involving some positive feedback at the favored pole seems simpler and therefore more likely.

We have considered the hypothesis that the two poles are competing for common precursors; however, we think that our results do not support this model. In a precursor competition model, it would be expected that when there is a decreased demand for precursors at the favorable pole, the rate of elongation would increase at the un-favored pole. In fact, we see in Figure 5A that, in different genetic backgrounds, growth can increase at one side of the side without compensatory decreases in growth on the other side (Δ*lamA*+ *pgfA*), and vice versa (Δ*lamA*+ *mmpL3*).

9) Line 215-219: Reference to TMM accumulation is made, but this comes well before the TMM levels are quantified later in the paper and the figure is not referenced. Some rewriting is needed to improve the flow of the paper here. Also, as mentioned above, I do not think the data showing TMM accumulation is convincing as presented.

See above.

10) Figure 2B: Please show the blots for the input material as well as the blots for anti-GFP as well as anti-FLAG blots.

Now included as Figure 2 —figure supplement 2.

11) Figure 2C: Make the font bigger for PgfA and MmpL3 or find some other way to make the labels clearer.

Labels for this figure have been made clearer.